# TAPBPR bridges UDP-glucose: glycoprotein glucosyltransferase 1 onto MHC class I to provide quality control in the antigen presentation pathway

Andreas Neerincx[1†], Clemens Hermann[1†‡], Robin Antrobus[2], Andy van Hateren[3,4], Huan Cao[5], Nico Trautwein[6], Stefan Stevanović[6], Tim Elliott[3,4], Janet E Deane[2], Louise H Boyle[1]*

[1]Department of Pathology, University of Cambridge, Cambridge, United Kingdom; [2]Cambridge Institute for Medical Research, University of Cambridge, Cambridge, United Kingdom; [3]Faculty of Medicine, University of Southampton, Southampton, United Kingdom; [4]Institute for Life Science, University of Southampton, Southampton, United Kingdom; [5]Division of Applied Medicine, Institute of Medical Sciences, University of Aberdeen, Aberdeen, United Kingdom; [6]Department of Immunology, Eberhard Karls University Tübingen, Tübingen, Germany

*For correspondence: lhb22@cam.ac.uk

[†]These authors contributed equally to this work

Present address: [‡]Department of Integrative Biomedical Sciences, Division of Chemical and Systems Biology, Institute for Infectious Disease and Molecular Medicine, University of Cape Town, Cape Town, South Africa

Competing interests: The authors declare that no competing interests exist.

**Abstract** Recently, we revealed that TAPBPR is a peptide exchange catalyst that is important for optimal peptide selection by MHC class I molecules. Here, we asked whether any other co-factors associate with TAPBPR, which would explain its effect on peptide selection. We identify an interaction between TAPBPR and UDP-glucose:glycoprotein glucosyltransferase 1 (UGT1), a folding sensor in the calnexin/calreticulin quality control cycle that is known to regenerate the $Glc_1Man_9GlcNAc_2$ moiety on glycoproteins. Our results suggest the formation of a multimeric complex, dependent on a conserved cysteine at position 94 in TAPBPR, in which TAPBPR promotes the association of UGT1 with peptide-receptive MHC class I molecules. We reveal that the interaction between TAPBPR and UGT1 facilities the reglucosylation of the glycan on MHC class I molecules, promoting their recognition by calreticulin. Our results suggest that in addition to being a peptide editor, TAPBPR improves peptide optimisation by promoting peptide-receptive MHC class I molecules to associate with the peptide-loading complex.

## Introduction

The presentation of antigenic peptides to the immune system by MHC class I molecules is crucial in generating protective responses against infection and cancer. Central to this process is the loading and optimisation of peptides onto MHC class I molecules within the peptide-loading complex (PLC) in the endoplasmic reticulum (ER) by tapasin, an MHC class I-dedicated chaperone that has been the focus of intense investigation for the past two decades (*Sadasivan et al., 1996*; *Ortmann et al., 1997*; *Williams et al., 2002*). It is now well established that tapasin functions as a peptide exchange catalyst for MHC class I molecules, a process that is important in the selection of high-affinity peptides onto MHC class I molecules (*Chen and Bouvier, 2007*; *Wearsch and Cresswell, 2007*). Recently, we revealed that TAPBPR, a second MHC class I-dedicated chaperone in the antigen presentation pathway, also functions as a peptide exchange catalyst for MHC class I molecules (*Boyle et al., 2013*; *Hermann et al., 2015*), a finding that was subsequently verified by Margulies and colleagues (*Morozov et al., 2016*). Thus, it is now clear that there are at least two MHC class

I-specific chaperones in the antigen presentation pathway that are intimately involved in selecting peptides for presentation on MHC class I molecules.

Although both tapasin and TAPBPR share the ability to optimise peptide selection in vitro, they cannot directly compensate for each other within a cellular environment and appear to influence peptide selection in separate yet complementary processes. In the absence of a functional tapasin molecule, inefficient peptide loading occurs, resulting in MHC class I molecules loaded with suboptimal peptide ligands (*Ortmann et al., 1997*; *Purcell et al., 2001*; *Williams et al., 2002*). As a consequence, the absence of tapasin produces thermolabile MHC class I complexes that are inefficiently expressed on the cell surface, although different MHC class I allomorphs differ in their dependency on tapasin (*Ortmann et al., 1997*; *Lewis et al., 1998*; *Peh et al., 1998*; *Garbi et al., 2000*; *Grandea et al., 2000*; *Williams et al., 2002*; *Rizvi et al., 2014*). In contrast to our understanding of tapasin, the precise role of TAPBPR-mediated peptide editing in the antigen presentation pathway has yet to be fully characterised (*Hermann et al., 2015*; *Morozov et al., 2016*). TAPBPR is not essential for the initial peptide-loading event onto MHC class I molecules (*Boyle et al., 2013*). Instead, TAPBPR has a more subtle, fine-tuning effect on the peptides displayed, removing some peptides of lower affinity and thus improving peptide selection and increasing the stability of MHC class I molecules (*Hermann et al., 2015*).

We have speculated that the different effects of tapasin and TAPBPR on the peptide repertoire in cells is due, at least in part, to the environment in which the two chaperones operate. Tapasin functions within the PLC in an environment that is rich in suitable peptides for MHC class I binding (*Sadasivan et al., 1996*; *Li et al., 1997*; *Ortmann et al., 1997*), which helps promote efficient peptide loading onto MHC class I molecules. In contrast TAPBPR is not a component of the PLC and therefore performs peptide editing outside this complex, potentially in a more peptide-deficient environment (*Boyle et al., 2013*; *Hermann et al., 2015*), which may favour peptide dissociation from MHC class I molecules. Therefore, it seems plausible that tapasin and TAPBPR have evolved to function in distinct cellular environments. For tapasin, three regions have been identified that are essential for its localisation and function within the PLC: its transmembrane domain is responsible for its interaction with TAP (*Petersen et al., 2005*; *Rufer et al., 2015*); a free cysteine residue at position C95 is essential for its association with ERp57 (*Dick et al., 2002*; *Peaper et al., 2005*); and residues in the Ig domains interact with MHC class I (*Turnquist et al., 2001*; *Turnquist et al., 2004*, *Dong et al., 2009*). For TAPBPR, the only functional sites to be identified so far are those that are responsible for its interaction with MHC class I (*Hermann et al., 2013*), and as yet, no association partners that function with TAPBPR have been characterised. Our aim here was to investigate whether any other co-factors interacted with TAPBPR in cells, which would explain the ability of TAPBPR to optimise peptide selection.

## Results

### TAPBPR binds to UDP-glucose:glycoprotein glucosyltransferase 1

To identify potential cellular binding partners for TAPBPR, we transiently transfected HeLaM cells with a construct in which the cDNA encoding amino acids 22–468 of human TAPBPR (i.e. the mature protein) was cloned downstream of a generic ER leader sequence, two protein A cassettes and a myc tag (ZZ-TAPBPR). This resulted in the expression of ZZ-TAPBPR at an approximately five-fold greater level than that observed for endogenous TAPBPR in IFN-γ-treated HeLaM. Affinity chromatography with IgG-sepharose beads was subsequently used to isolate ZZ-TAPBPR and any associated proteins. As expected, tandem mass spectrometry (MS/MS) analysis identified the MHC class I heavy chain and $\beta$2m in the ZZ-TAPBPR immunoprecipitate (*Table 1*) (*Boyle et al., 2013*). Interestingly, UDP-glucose:glycoprotein glucosyltransferase 1 (UGT1) was also isolated in the immunoprecipitates from ZZ-TAPBPR-expressing cells, but not in control HeLaM cells transfected with an empty vector (*Table 1*), suggesting that UGT1 could be a novel binding partner for TAPBPR. UGT1 was ranked as the third specific hit in the ZZ-TAPBPR pulldown after the MHC class I heavy chain and TAPBPR (*Table 1*). UGT1 is an ER/cis-Golgi resident enzyme that monitors glycoprotein folding (*Arnold et al., 2000*; *Tessier et al., 2000*; *Zuber et al., 2001*; *D'Alessio et al., 2010*). In 2011, Cresswell and colleagues showed that UGT1 plays an important role in the MHC class I antigen processing and presentation pathway (*Wearsch et al., 2011*; *Zhang et al., 2011*). Although no direct

**Table 1.** Selected proteins identified in IgG-sepharose pulldowns on ZZ-TAPBPR

Affinity chromatography with IgG-sepharose was performed on HeLaM cells expressing a protein-A-tagged TAPBPR molecule (ZZ-TAPBPR) or HeLaM cells transduced with an empty vector (control). Immunoprecipitates were analysed by in gel tryptic digest followed by liquid chromatography-tandem mass spectrometry and data were processed using Scaffold. Identified proteins are shown with their exclusive unique peptide count, percentage coverage, and exclusive unique spectrum count as determined by Scaffold. Rank denotes the position when data are sorted by exclusive unique peptide count with all proteins present in the control removed. Pep: exclusive unique peptide count; Cov: percentage coverage; Count: exclusive unique spectrum count.

| Protein | Gene name | Control | | ZZ-TAPBPR | | Rank |
| --- | --- | --- | --- | --- | --- | --- |
| | | Pep (Cov) | Count | Pep (Cov) | Count | |
| Tapasin-related protein | TAPBPL | – | – | 8 (16) | 11 | 2 |
| HLA class 1, A-68 | HLA-A | – | – | 14 (35) | 21 | 1 |
| β-2-microglobulin | β2M | – | – | 1 (8.4) | 1 | 95 |
| UDP-glucose:glycoprotein glucosyltransferase 1 | UGGT1 | – | – | 10 (7.3) | 10 | 3 |

association was demonstrated between MHC class I and UGT1 within a cellular environment, they found that MHC class I maturation and assembly was delayed, surface expression of MHC class I was reduced, and there was an impairment of peptide selection in UGT1-deficient cells (*Zhang et al., 2011*). Furthermore, recombinant UGT1 was found to reglucosylate MHC class I molecules associated with suboptimal ligands and permitted their re-engagement with the PLC, thus providing direct evidence for the role of UGT1 in the antigen presentation pathway (*Wearsch et al., 2011*). To confirm that the interaction between TAPBPR and UGT1 was not an artefact of tagging or TAPBPR overexpression, immunoprecipitation of TAPBPR was performed in IFN-γ-induced HeLaM and KBM-7 cells, followed by western blotting for UGT1. An association between endogenous TAPBPR and

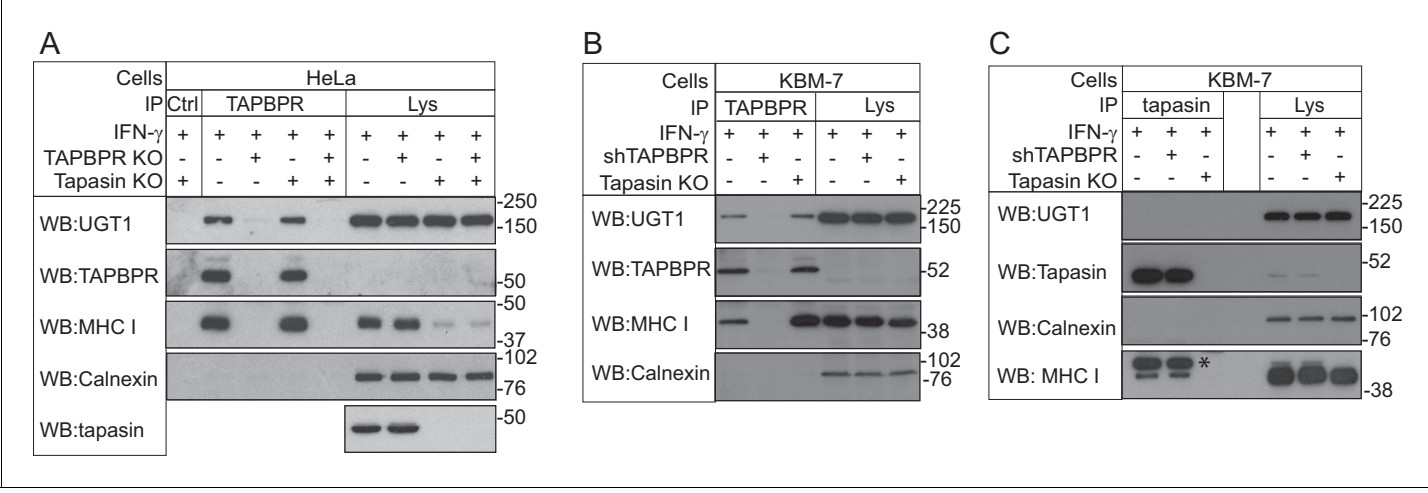

**Figure 1.** TAPBPR associates with UDP-glucose:glycoprotein glucosyltransferase 1 (**A** and **B**) TAPBPR or (**C**) tapasin were immunoprecipitated using PeTe4 or Pasta1, respectively, from (**A**) IFN-γ-treated HeLaM, TAPBPR KO, tapasin KO, and double tapasin KO/TAPBPR KO HeLaM cells or (**B** and **C**) IFN-γ-treated wild-type, TAPBPR-depleted (+ shTAPBPR), and tapasin KO KBM-7 cells. In (**A**), immunoprecipitation with an isotype control antibody was used as a control (labelled 'Ctrl'). Western blot analysis was performed for UGT1, TAPBPR (using R021), tapasin (Rgp48N), MHC class I HC (using 3B10.7), or calnexin as a loading control on immunoprecipitates or lysates as indicated. Note: the amounts of TAPBPR present in the lysates were below limits of detection and were detectable after concentration via immunoprecipitation. In (**C**), * denotes the tapasin protein from the previous blot, while the lower band is the MHC class I HC. The data shown are representative of three independent experiments. In (**A** and **B**), an increased association between TAPBPR and MHC class I molecules in the absence of tapasin can be observed. While this is not immediately obvious in (**A**) with HeLaM cells, a significant loss of MHC class I expression is observed in the absence of tapasin in this cell line. Therefore, a relative increase in the association between MHC class I molecules and TAPBPR is observed in the absence of tapasin. KO: knockout; UGT1: UDP-glucose:glycoprotein glucosyltransferase 1; IP: immunoprecipitation; WB: western blot.

UGT1 was observed in both IFN-γ-treated HeLaM and KBM-7 cells (*Figure 1A and B* respectively). The association was not observed in cells that were knocked out (*Figure 1A*) or depleted (*Figure 1B*) of TAPBPR, demonstrating that the co-precipitation of UGT1 was a direct consequence of TAPBPR presence. Furthermore, the association between TAPBPR and UGT1 was observed in the absence of tapasin (*Figure 1A and B*), suggesting that a functional PLC is not required for TAPBPR to associate with UGT1. As shown in *Figure 1A and B*, we observed that the association between TAPBPR and MHC class I was increased in the absence of tapasin, an observation that is consistent with our previously published findings (*Hermann et al., 2013*). In contrast to the association observed between TAPBPR and UGT1, no association was observed between tapasin and UGT1 in IFN-γ-induced KBM-7 cells (*Figure 1C*). These results confirm that UGT1 is a novel binding partner for TAPBPR. However, as human TAPBPR lacks an N-linked glycan and UGT1 monitors glycoprotein folding by recognising hydrophobic patches near a $Man_9GlcNAc_2$ moiety (*Trombetta et al., 1989*; *Caramelo et al., 2003*, *Caramelo et al., 2004*; *Ritter et al., 2005*), it is highly unlikely that UGT1 functions directly in the quality control of TAPBPR.

## Residue C94 in TAPBPR is not involved in an intramolecular disulphide bond

Since the extracellular domain of TAPBPR contains seven cysteine residues (*Teng et al., 2002*), we were intrigued as to the probable existence, and functional relevance, of an unpaired cysteine residue. For tapasin, intramolecular disulphide bonds exist between C7 and C71 in the N-terminal domain and C295 and C363 in the membrane proximal Ig-like domain, while C95 is known not to be involved in an intramolecular disulphide bound, but instead forms an intermolecular disulphide bond with C57 of ERp57 (*Herberg et al., 1998*; *Dick et al., 2002*; *Dong et al., 2009*)(*Figure 2A and B*). Our Fold and Functional Assignment System (FFAS) model of TAPBPR (*Hermann et al., 2013*) now supported by small-angle X-ray scattering (SAXS) data (*Morozov et al., 2016*), predicts that C18 and C101 in the N-terminal domain and C300 and C361 in the IgC domain of TAPBPR form similar intramolecular disulphide bonds as those found in tapasin, and that C191 and C262 form an additional intramolecular disulphide bond (*Figure 2C*). This analysis predicts that C94 in TAPBPR is very unlikely to be involved in an intramolecular disulphide bond (*Figure 2C*). To verify this, all cysteine residues were individually changed to alanine, cloned into a lentiviral vector containing a bicistronic GFP, and then successfully transduced into HeLaM cells (*Figure 2—figure supplement 1*). Mutation of C300 or C361 in the IgC domain led to significantly reduced steady-state expression of TAPBPR (*Figure 2D*), suggesting that these cysteines are essential for TAPBPR stability. Under non-reducing conditions, TAPBPR^C18A and TAPBPR^C101A exhibited lower electrophoretic mobility relative to TAPBPR^WT (*Figure 2D*), suggesting that a disulphide bond exists between C18 and C101 that is important for the structural integrity of the N-terminal domain. Mutation of C191 and C262 did not significantly affect steady-state TAPBPR expression; however, a very subtle change in electrophoretic mobility of these two mutants was observed under non-reducing conditions compared to TAPBPR^WT (*Figure 2D*). This may reflect disruption of a disulphide bond within a stable protein domain that does not lead to sufficient unfolding as to be detected by electrophoresis, as is the case for MHC class II molecules (*Kaufman and Strominger, 1982*, *1983*). Another possibility is that, under the conditions used, the sample was reoxidised prior to loading. In contrast, mutation of C94 in TAPBPR did not affect steady-state protein expression or electrophoretic mobility (*Figure 2D*). As an additional check to determine whether TAPBPR^C94A was stable and folded, this mutant was cloned into the pHLsec expression vector and transiently transfected into HEK293F cells. As observed with TAPBPR^WT (*Figure 2E*) (*Hermann et al., 2015*), TAPBPR^C94A was efficiently expressed and purified using Ni affinity and size exclusion chromatography (*Figure 2F*). Using differential scanning fluorimetry, the melting temperature of purified TAPBPR^WT and TAPBPR^C94A were the same, demonstrating that mutation of C94 to alanine did not affect protein stability (*Figure 2G*).

## C94 in TAPBPR is essential for its association with UGT1

To explore the functional importance of the unpaired cysteine at position 94, we reconstituted the TAPBPR-knockout HeLaM cell line (HeLaM^KO) we characterised previously (*Hermann et al., 2015*) with either TAPBPR^WT or TAPBPR^C94A. First, we asked whether the interactions between TAPBPR and other proteins were altered when C94 was mutated to alanine. Comparison of TAPBPR

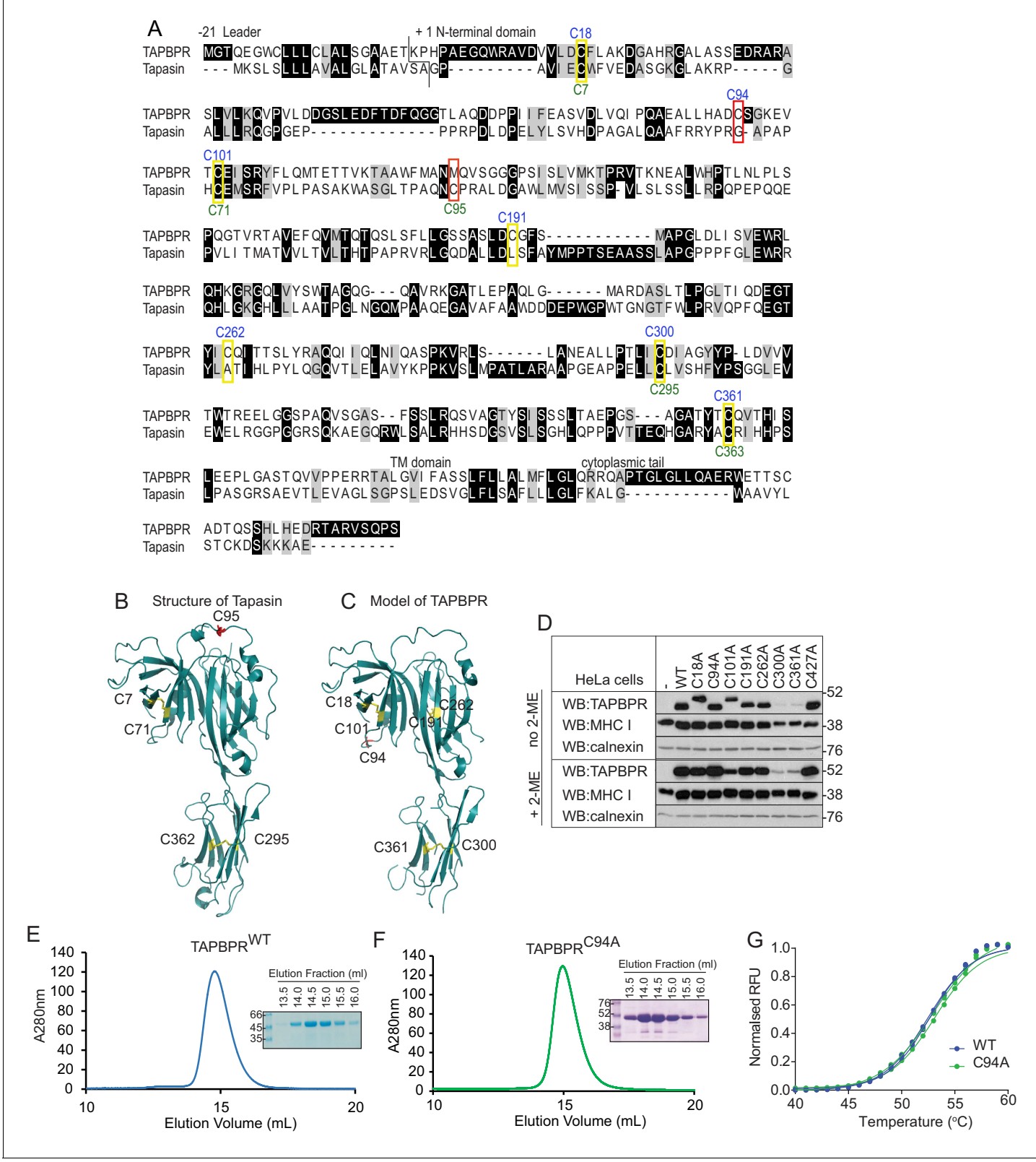

**Figure 2.** C94 in TAPBPR is not involved in an intramolecular disulphide bond (**A**) Amino acid sequences of human TAPBPR (NP_060479.3) and tapasin (AAC20076.1) were aligned using ClustalW. Cysteine residues are marked in yellow boxes, with the cysteine in tapasin that interacts with ERp57 and the predicted unpaired cysteine in TAPBPR highlighted in red. The positions of the cysteine residues in TAPBPR and tapasin are labelled above and below, in blue or green, respectively. (**B**) Structure of tapasin (Protein Data Bank ID: 3F8U), with the cysteines involved in intramolecular disulphide bonds

*Figure 2 continued on next page*

*Figure 2 continued*

highlighted in yellow and the free cysteine C95 highlighted in red. (C) FFAS model for TAPBPR (*Hermann et al., 2013*), with potential disulphide bridges highlighted in yellow and the predicted free cysteine C94 highlighted in red. (D) Lysates from a HeLaM cell panel expressing cysteine-mutant TAPBPR molecules were resolved under non-reducing (no 2-ME) or reducing (+2-ME) conditions, then blotted for TAPBPR (using mouse anti-TAPBPR), MHC class I heavy chain (using HC10), or calnexin as a loading control. The antibody used to detect TAPBPR was raised against the membrane distal domain (aa 23–122); therefore, it is unlikely that the lack of detection of C300 and C361 is due to a lack of antibody recognition of these IgC domain mutants. Data shown in (D) are representative of three independent experiments. (E and F) Size exclusion chromatogram of TAPBPR$^{WT}$ and TAPBPR$^{C94A}$ purified from cell culture supernatant. The protein peaks were analysed by SDS-PAGE followed by Coomassie staining. (G) Differential scanning fluorimetry of TAPBPR$^{WT}$ and TAPBPR$^{C94A}$ demonstrates equivalent thermal denaturation profiles. WT: wild-type; WB: western blot; RFU: relative fluorescence units.

The following figure supplement is available for figure 2:

**Figure supplement 1.** Transduction efficiency of the cysteine-mutant panel into HeLaM cells

immunoprecipitates from HeLaM$^{KO}$TAPBPR$^{WT}$ and HeLaM$^{KO}$TAPBPR$^{C94A}$ by MS/MS analysis revealed that TAPBPR still associated with both the MHC class I heavy chain and $\beta$2m in the absence of C94 (*Table 2*). However, the most striking difference between the immunoprecipitates was the absence of UGT1 in the HeLaM$^{KO}$TAPBPR$^{C94A}$ sample (*Table 2*). These findings were confirmed by western blot analysis on TAPBPR immunoprecipitates, where the amount of UGT1 that was co-immunoprecipitated was drastically reduced following mutation of C94 (*Figure 3A*). These results suggest that the C94 residue in TAPBPR is important for its association with UGT1. Given that all lumenal cysteine residues are conserved in TAPBPR from different species (*Figure 3—figure supplement 1*), it is possible that the association between TAPBPR and UGT1 is conserved from humans to fish.

## TAPBPR does not form a disulphide-linked dimer with UGT1

Having revealed that the association between TAPBPR and UGT1 occurred in a C94-dependent manner, we tested whether a disulphide bond formed between the two proteins. Under non-reducing conditions, there was no evidence of a disulphide-linked heterodimer between TAPBPR and UGT1 (*Figure 3B and C*). Both UGT1 and TAPBPR isolated from TAPBPR immunoprecipitates from HeLaM$^{KO}$TAPBPR$^{WT}$ cells resolved at the predicted size of the monomeric proteins (~175 kDa for UGT1 and <52 kDa for TAPBPR) (*Figure 3B*). Furthermore, no difference in electrophoretic mobility was observed if the cells expressed TAPBPR$^{WT}$ or TAPBPR$^{C94A}$ (*Figure 3B*). In addition to suggesting that TAPBPR does not form a disulphide-linked dimer with UGT1, the results also indicate that TAPBPR does not form a covalent bond with any other as-yet unidentified binding partner of measurable size in these cells. There was also no evidence of a disulphide-linked dimer between endogenously expressed TAPBPR and UGT1 in IFN-γ-induced KBM-7 cells (*Figure 3C*). Based on our FFAS model of TAPBPR (*Hermann et al., 2013*), additional residues (I83, E87, L90, H91, and D93)

**Table 2.** Selected proteins identified in TAPBPR co-immunoprecipitates

TAPBPR was immunoprecipitated using PeTe4 from IFN-γ-treated HeLaM-TAPBPR$^{KO}$(HeLaM$^{KO}$) cells reconstituted with either TAPBPR$^{WT}$ or TAPBPR$^{C94A}$. Immunoprecipitates were analysed by in gel tryptic digest followed by liquid chromatography-tandem mass spectrometry and data were processed using Scaffold. Identified proteins are shown with their exclusive unique peptide count, total percentage coverage, and exclusive unique spectrum count as determined by Scaffold. Pep: exclusive unique peptide count; Cov: percentage coverage; Count: exclusive unique spectrum count.

| Protein | Gene name | TAPBPR$^{WT}$ | | TAPBPR$^{C94A}$ | |
|---|---|---|---|---|---|
| | | Pep (Cov) | Count | Pep (Cov) | Count |
| Tapasin-related protein | *TAPBPL* | 32 (43) | 54 | 29 (47) | 50 |
| HLA class 1, A-68 | *HLA-A* | 50 (64) | 88 | 41 (59) | 70 |
| $\beta$-2-microglobulin | *$\beta$2M* | 4 (46) | 7 | 1 (8.4) | 1 |
| UDP-glucose:glycoprotein glucosyltransferase 1 | *UGGT1* | 19 (11) | 25 | – | – |

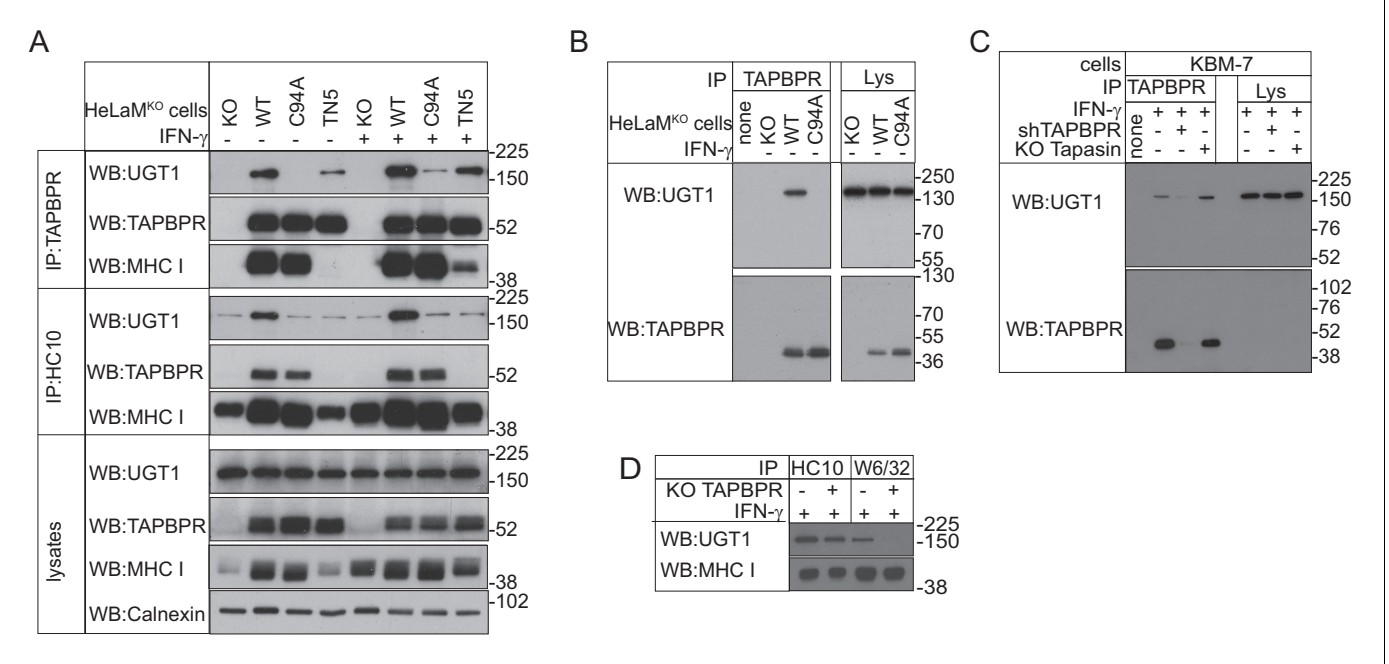

**Figure 3.** TAPBPR binds to UDP-glucose:glycoprotein glucosyltransferase 1 in a C94-dependent manner, but not via a disulphide bond and bridges UDP-glucose:glycoprotein glucosyltransferase 1 to MHC class I molecules (A–C) TAPBPR or (A) HC10-reactive MHC class I were isolated by immunoprecipitation from (A and B) HeLaM-TAPBPR^KO cells (HeLaM^KO) and HeLaM^KO cells reconstituted with either TAPBPR^WT, TAPBPR^C94A, or TAPBPR^TN5 or (C) WT, TAPBPR-depleted (+ shTAPBPR), and tapasin KO KBM-7 cells, with or without IFN-γ treatment, as indicated under (A) reducing or (B and C) non-reducing conditions. (D) HC10 and W6/32-reactive MHC class I molecules were isolated from IFN-γ-treated HeLaM and HeLaM^KO cells and resolved under reducing conditions. Western blot analysis was performed for UGT1, TAPBPR (using R021), MHC class I HC (using 3B10.7), or calnexin as a loading control on immunoprecipitates or lysates as indicated. The lane labelled 'none' in (B) indicates PeTe4 antibody only, with no cellular lysate in immunoprecipitation. All data shown are representative of three independent experiments. KO: knockout; UGT1: UDP-glucose: glycoprotein glucosyltransferase 1; WT: wild-type; WB: western blot; IP: immunoprecipitation.

The following figure supplements are available for figure 3:

**Figure supplement 1.** Cysteine residues are conserved in TAPBPR across different species

**Figure supplement 2.** Residues in the helix next to C94 in TAPBPR influence UDP-glucose:glycoprotein glucosyltransferase 1 association

predicted to form an α-helix on the surface of TAPBPR near C94 were also anticipated to contribute to the UGT1 interface. Mutation of these residues (UBS1: I83K, E87K, or UBS2: E87K, L90K, H91S, and D93R) decreased the association of UGT1 with TAPBPR (*Figure 3—figure supplement 2*). This suggests that the interaction of TAPBPR with UGT1 may involve an extended interface.

## TAPBPR acts as a bridge between UGT1 and MHC class I heterodimers

Our results thus far suggest that distinct regions of TAPBPR are responsible for binding its interaction partners, with residues in IgV and IgC domains being critical for binding MHC class I molecules (*Hermann et al., 2013*) and C94 in the N-terminal domain being important for associating with UGT1. Next, we sought to determine whether a tri-molecular complex between TAPBPR, UGT1, and MHC class I formed in cells by turning our attention to the contribution of MHC class I to the interactions. First, we asked whether an association between TAPBPR and MHC class I was required for UGT1 to efficiently associate with TAPBPR. In TAPBPR immunoprecipitates from HeLaM^KO cells transduced with TAPBPR^TN5, which is unable to bind to MHC class I molecules (*Hermann et al., 2013*), the amount of UGT1 bound to TAPBPR was significantly reduced compared to cells expressing TAPBPR^WT (*Figure 3A*). Interestingly, upon IFN-γ treatment, which boosts MHC class I expression, TAPBPR^TN5 was able to interact with MHC class I molecules, albeit weakly, and the amount of

UGT1 bound to TAPBPR[TN5] similarly increased (*Figure 3A*). Together, these results suggest that the association between TAPBPR and MHC class I is important for the maximum association of UGT1 with TAPBPR, supporting the concept that a tri-molecular complex forms between TAPBPR, UGT1, and MHC class I.

We wondered whether TAPBPR was acting as a molecular platform or intermediate between UGT1 and MHC class I, which is likely to be peptide free or loaded with low-affinity peptides, given recent findings (*Hermann et al., 2015*; *Morozov et al., 2016*). To test this, we determined whether TAPBPR was required for UGT1 to associate with MHC class I. In HeLaM[KO] cells, we observed a weak association between MHC class I and UGT1 (*Figure 3A*). However, in HeLaM[KO]TAPBPR[WT] cells, this association was significantly enhanced (*Figure 3A*). No such enhancement was observed in cells expressing the mutant forms TAPBPR[C94A] or TAPBPR[TN5], even in the presence of IFN-γ (*Figure 3A*). This suggests that the association between MHC class I molecules and UGT1 is promoted by TAPBPR. We also tested the requirement for TAPBPR in the association between MHC class I molecules and UGT1 when endogenous TAPBPR was induced in HeLaM cells by IFN-γ. UGT1 was found to associate weakly with both HC10- and W6/32-reactive MHC class I molecules in IFN-γ-treated HeLaM cells (*Figure 3D*). However, in IFN-γ-treated HeLaM[KO] cells, a significant reduction was observed between UGT1- and W6/32-reactive MHC class I molecules (*Figure 3D*). These results are consistent with TAPBPR acting as a bridge between UGT1 and MHC class I heterodimers and suggest that UGT1 recognition of MHC class I molecules can occur in a TAPBPR-independent manner (i.e. for the majority of HC10-reactive MHC class I) or in a TAPBPR-dependent manner (i.e with W6/32-reactive MHC class).

## TAPBPR[C94A] still functions as a peptide editor in vitro

Recently, we and others have shown that TAPBPR is a peptide exchange catalyst that is important for optimal peptide selection by MHC class I molecules (*Hermann et al., 2015*; *Morozov et al., 2016*). Having identified an interaction between UGT1 and TAPBPR, we next asked how important this association was for peptide optimisation on MHC class I molecules in a cellular environment. First, we tested whether TAPBPR[C94A] was a functionally active MHC class I peptide editor in vitro. Using fluorescence polarisation, we observed a comparable ability of TAPBPR[C94A] and TAPBPR[WT] to enhance peptide dissociation and association on HLA-A*02:01 in vitro (*Figure 4A and B*). Furthermore, when we tested the ability of TAPBPR[C94A] to discriminate between high-affinity (the labelled peptide FLPSDC*FPSV) and lower-affinity (NLVPMVATV) peptides for binding to HLA-A*02:01 (see *Figure 4—figure supplement 1* for a comparison of the affinity of these two peptides for HLA-A2), in the presence of TAPBPR[C94A], NLVPMVATV became a poorer competitor, as observed with TAPBPR[WT] (*Figure 4C*) (*Hermann et al., 2015*). These results suggest that the peptide-editing function of TAPBPR is not altered as a direct consequence of mutation of the C94 residue.

## UGT1 bound to TAPBPR helps improve peptide selection on MHC class I molecules

When we compared the surface expression of MHC class I molecules on HeLaM[KO] cells reconstituted with TAPBPR[C94A] or TAPBPR[WT], we found that both molecules downregulated total MHC class I, as detected by W6/32, to a similar extent (*Figure 5A*). However, we observed slightly lower surface expression of HLA-A*68:02 in cells expressing TAPBPR[C94A] compared to TAPBPR[WT] (*Figure 5A and B*) despite comparable levels of steady-state expression of TAPBPR in cell lines transduced with TAPBPR[WT] and TAPBPR[C94A] (*Figure 3A*). Next, in order to determine the contribution that the UGT1 bound to TAPBPR has on peptide selection in a cellular environment, we compared the MHC class I peptide repertoire on HeLaM[KO]TAPBPR[WT] with HeLaM[KO]TAPBPR[C94A] cells, both with the addition of IFN-γ to boost MHC class I, TAP, and tapasin expression. In this system, both cell types still contain functional UGT1 molecules and express TAPBPR molecules that remain capable of peptide exchange, as shown above. Therefore, it allows for the loss of the association between TAPBPR and UGT1 to be specifically examined. Comparison of the amino acid sequences of peptides eluted from MHC class I molecules isolated from cells expressing TAPBPR[WT] and TAPBPR[C94A] by mass spectrometry revealed differences in the repertoire of peptides presented. A total of 690 peptides were shared between these two cell lines, while 500 were unique to HeLaM[KO]TAPBPR[WT] cells and 174 were unique to HeLaM[KO]TAPBPR[C94A] cells (*Figure 5C*). Comparison of the anchor residues

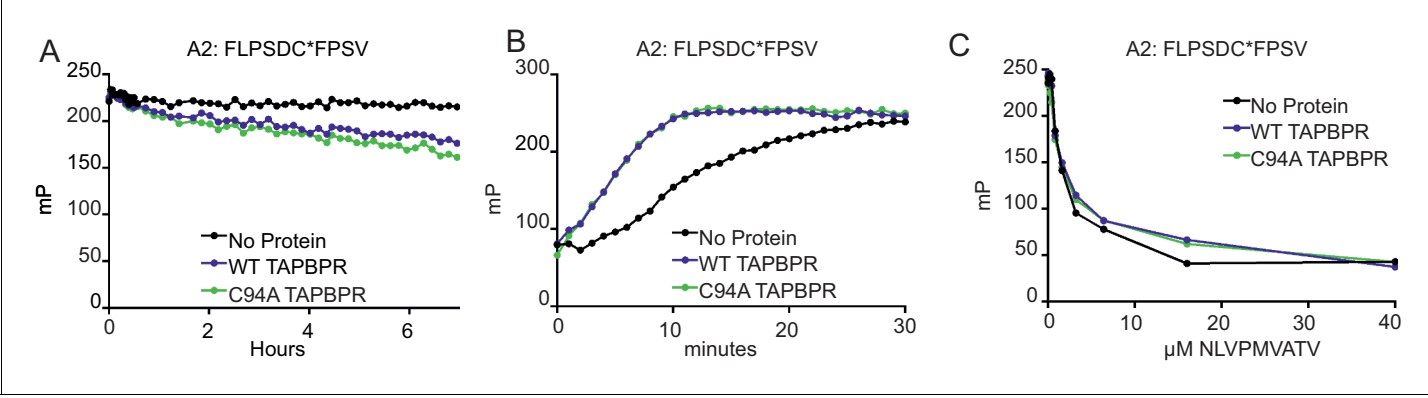

**Figure 4.** TAPBPR[C94A] still functions as a peptide editor in vitro (**A**) Dissociation of the fluorescent peptide FLPSDC*FPSV from HLA-A*02:01 in the absence or presence of TAPBPR[WT] or TAPBPR[C94A]. 500 nM HLA-A02:01 molecules (refolded with UV-labile KILGFVFjV peptide, as described previously in *Hermann et al., 2015*) were exposed to 366 nm UV light at 4°C for 20 min. The UV-exposed protein was then incubated with 17.6 nM FLPSDC*FPSV (C* denotes 5-carboxytetramethylrhodamine 'TAMRA'-labelled cysteine) overnight at room temperature. The FLPSDC*FPSV–HLA-A*02:01 complexes were then split and incubated with 1000-fold molar excess NLVPMVATV with either buffer (no protein) or supplemented with either 0.125 μM TAPBPR[WT] or TAPBPR[C94A]. The data shown are representative of three experiments. (**B**) Association of the fluorescent peptide FLPSDC*FPSV with HLA-A*02:01 in the absence or presence of TAPBPR[WT] or TAPBPR[C94A]. 75 nM HLA-A02:01fos molecules (refolded with UV-labile KILGFVFjV peptide) were mixed with 1.5 μM human β2m and exposed to 366-nm UV light at 4°C for 20 min, and then 5.95 nM FLPSDC*FPSV was added in the absence or presence of 0.125 μM TAPBPR[WT] or TAPBPR[C94A] immediately before fluorescence polarisation measurements were taken. One representative experiment of three is shown. (**C**) Competition between peptides for binding with peptide-receptive HLA-A*02:01 molecules in the presence or absence of TAPBPR[WT] or TAPBPR[C94A] as measured by fluorescence polarisation. 75 nM HLA-A02:01fos molecules (refolded with UV-labile KILGFVFjV peptide) were mixed with 1.5 μM human β2m and exposed to 366-nm UV light at 4°C for 20 min, and then incubated with 5.95 nM high-affinity peptide FLPSDC*FPSV and various concentrations of the lower-affinity competing peptide NLVPMVATV (0–100 μM) in the presence or absence of 0.0625 μM TAPBPR[WT] or TAPBPR[C94A] (see *Figure 4—figure supplement 1* for a comparison of the affinity of FLPSDCFPSV or NLVPMVATV for HLA-A2). Fluorescence polarisation measurements were taken after incubation overnight at 25°C. One representative experiment of three is shown. WT: wild-type.

The following figure supplement is available for figure 4:

**Figure supplement 1.** Comparison of the ability of FLPSDCFPSV or NLVPMVATV to inhibit binding of FLPSDC*FPSV to peptide-receptive HLA-A*02:01fos molecules

found at the P2 and C-terminal (PΩ) positions of the peptides unique to TAPBPR[WT] cells (presumably peptides selected by the TAPBPR:UGT1 complex) with peptides unique to TAPBPR[C94A] cells (presumably peptides restricted by the TAPBPR:UGT1 complex) revealed a decrease in the canonical anchor residues in the peptides uniquely isolated from the TAPBPR[C94A]-expressing cells for HLA-A*68:02 (*Figure 5D*). For HLA-B*15:03, no decrease in the prevalence of anchor residues was observed (*Figure 5D*). These results suggest that the ability of TAPBPR to improve peptide optimisation on MHC class I molecules within a cellular environment can be strongly influenced by its association with UGT1, as observed here for HLA-A*68:02.

## Cells expressing TAPBPR[C94A] have increased surface expression of peptide-receptive HLA-A68

We also independently verified the presence of suboptimally loaded HLA-A68 on the surface of cells expressing TAPBPR[C94A] compared to TAPBPR[WT] by comparing HLA-A68 levels on cells in the absence and presence of exogenously added high-affinity peptides for MHC class I molecules. It is noteworthy that these particular experiments have been performed in independently produced cell lines in the unmodified HeLaM background (i.e. without TAPBPR[KO], in the absence of IFN-γ induction, and with lower transgene expression) (see *Figure 5—figure supplement 4* for the association of TAPBPR with MHC class I and UGT1 in these cell lines). On HeLaM-TAPBPR[WT] cells, only a slight increase in HLA-A68 expression was observed upon incubation at 26°C with exogenous peptide compared to cells incubated at 37°C without peptide (*Figure 5E and F*). In contrast, on HeLaM-TAPBPR[C94A] cells, a significant increase in HLA-A68 expression was observed on cells incubated at

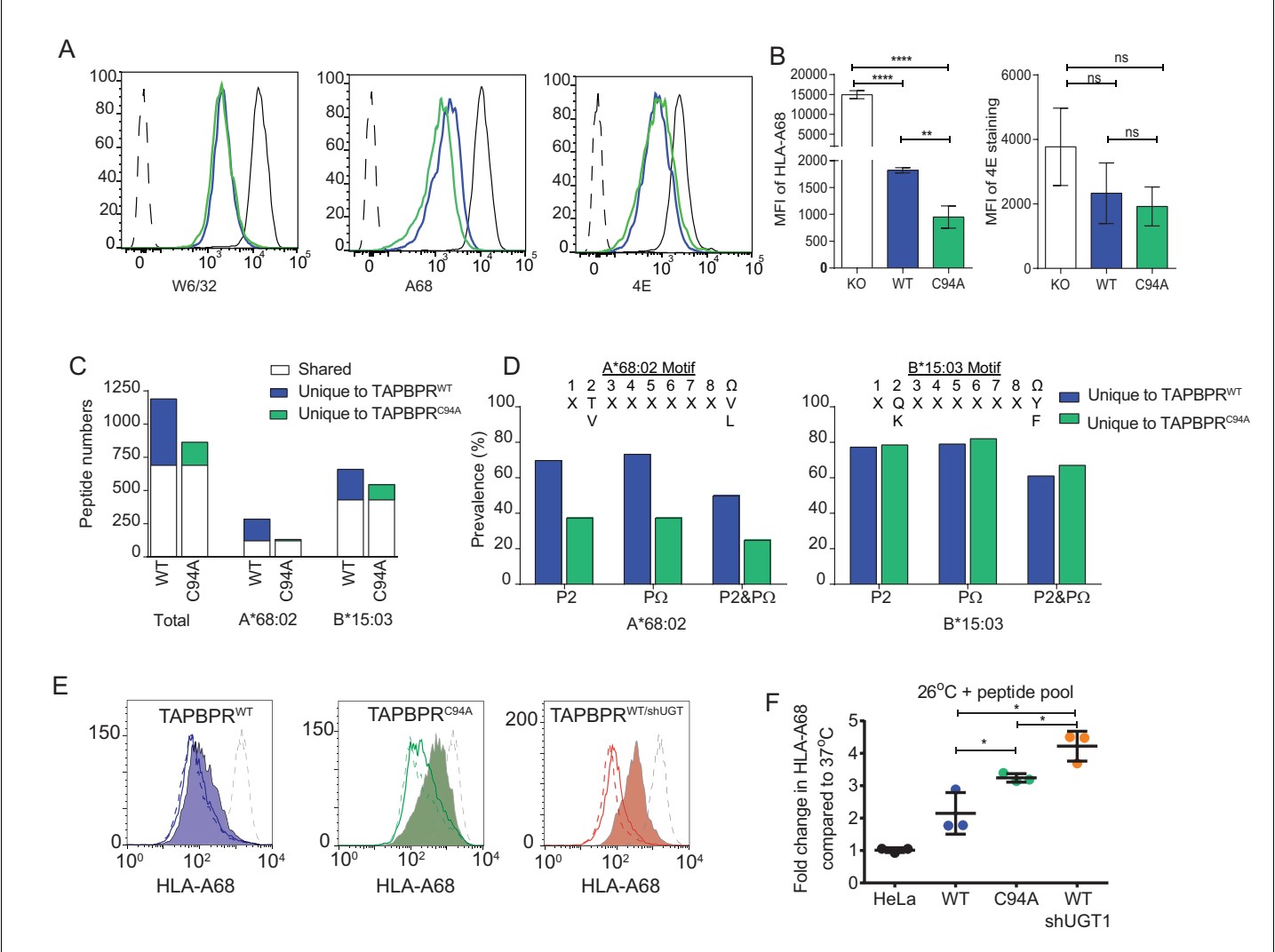

**Figure 5.** UDP-glucose:glycoprotein glucosyltransferase 1 bound to TAPBPR influences peptide selection on MHC class I molecules (A) Cytofluorimetric analysis of W6/32-, HLA-A68-, and 4E-reactive MHC class I molecules on HeLaM$^{KO}$cells (black line), HeLaM$^{KO}$cells reconstituted with TAPBPR$^{WT}$ (blue line) or TAPBPR$^{C94A}$cells (green line). Staining of HeLaM$^{KO}$ cells with an isotype control is included as a negative control (black dashed histogram). (B) Bar graphs show the MFI for cell surface HLA-A68 and for HLA-B molecules using 4E antibody from three independent experiments, as performed in (A). Error bars represent SD. ****p<0.0001, **p<0.005, ns=not significant, based on unpaired t tests. The MFI for cell surface HLA-A68 on IFN-γ-treated equivalents can be seen in *Figure 5—figure supplement 1*. (C and D) Peptide–MHC class I complexes were isolated by affinity chromatography using W6/32 from IFN-γ-treated HeLaM$^{KO}$, HeLaM$^{KO}$TAPBPR$^{WT}$, and HeLaM$^{KO}$TAPBPR$^{C94A}$ cells. Eluted peptides were analysed using liquid chromatography tandem mass spectrometry (for full list see *Figure 5—source data 1*). (C) Bar graph displaying the number of unique and shared peptides found in IFN-γ treated HeLaM$^{KO}$TAPBPR$^{WT}$ and HeLaM$^{KO}$TAPBPR$^{C94A}$ cells. (D) Graphs showing the prevalence of classic peptide anchors on HLA-A*68:02 (T and V at P2, V and L at PΩ) and HLA-B*15:03 (Q and K at P2, Y and F at PΩ) for peptides unique to IFN-γ treated TAPBPR$^{WT}$-expressing cells (i.e. peptides presumably permitted release by UGT1 in the context of TAPBPR) and peptides unique to IFN-γ treated TAPBPR$^{C94A}$-expressing cells (i.e. peptide presumably restricted by TAPBPR:UGT1) after removal of peptides shared with IFN-γ induced HeLaM$^{KO}$ cells. The data in (C and D) were generated from tandem mass spectrometry analysis performed three times on one immunoprecipitate. WebLogo depictions of the peptide sequences of 9-mers isolated from TAPBPR$^{WT}$- and TAPBPR$^{C94A}$-expressing cells can be found in *Figure 5—figure supplement 2*. Further statistical analysis on the isolated peptides can be found in *Figure 5—figure supplement 3*. (E) Cytofluorimetric analysis of peptide-loaded HLA-A68 on HeLaM-TAPBPR$^{WT}$, HeLaM-TAPBPR$^{C94A}$, and shUGT1-depleted HeLaM-TAPBPR$^{WT}$cells after incubation at 37°C (dashed line), 26°C (solid line), or 26°C supplemented with a pool of high-affinity peptides for HLA-A and -B (an influenza virus, Epstein-Barr virus, and cytomegalovirus [FEC] peptide pool. Individual peptides were used at a final concentration of 3 μg/ml) (filled histogram) for 90 min in media post-trypsinisation. HLA-A68 staining on non-transduced HeLaM cells incubated at 37°C for 90 min post-trypsinisation was included as a positive control (grey dashed line). y-axes = events normalised to mode. See *Figure 5—figure supplement 4* for associations between TAPBPR, MHC class I, and UGT1 in these independently produced cell lines. (F) Scatter dot plot show the fold change in MFI for cell surface HLA-A68 on cells at 26°C plus FEC peptide pool compared to incubation at 37°C alone on three

*Figure 5 continued on next page*

Figure 5 continued

independent replicates as performed in (E). Error bars represent SD. *p<0.05, based on unpaired t tests. MFI: median fluorescence intensity; KO: knockout; WT: wild-type; UGT1: UDP-glucose:glycoprotein glucosyltransferase 1.

The following source data and figure supplements are available for figure 5:

Source data 1. MHC class I peptide elution from IFN-γ treated HeLaM$^{KO}$TAPBPR$^{WT}$ and HeLaM$^{KO}$TAPBPR$^{C94A}$

Figure supplement 1. Surface expression of HLA-A68 upon IFN-γ treatment

Figure supplement 2. WebLogo depictions of the peptide sequences of 9-mers isolated from TAPBPR$^{WT}$- and TAPBPR$^{C94A}$-expressing cells WT: wild-type.

Figure supplement 3. Statistical analysis of peptides isolated from cells expressing TAPBPR$^{WT}$ and TAPBPR$^{C94A}$

Figure supplement 4. Associations between TAPBPR, MHC class I, and UDP-glucose:glycoprotein glucosyltransferase 1 in the HeLaM cell lines used to test peptide receptivity of HLA-A68

26°C with exogenous peptide compared to 37°C without peptide (*Figure 5E and F*). To further confirm that the increased expression of peptide-receptive HLA-A68 at the cell surface was a direct consequence of the loss of the association between TAPBPR and UGT1, as opposed to any other consequential effect of mutation of the C94 residue of TAPBPR, we depleted UGT1 in HeLaM-TAPBPR$^{WT}$ cells. In shUGT1-depleted cells, a significant, although not complete, reduction in the association between TAPBPR$^{WT}$ and UGT1 was observed, and the strong association between TAPBPR and MHC class I was maintained (*Figure 5—figure supplement 4*). However, the surface phenotype of HLA-A68 molecules now resembled that of HeLaM-TAPBPR$^{C94A}$ cells rather than that of HeLaM-TAPBPR$^{WT}$ cells, with increased levels of peptide-receptive HLA-A68 that were stabilised upon incubation at 26°C with high-affinity peptide (*Figure 5E and F*). The expression of peptide-receptive HLA-A68 was higher on HeLaM-TAPBPR$^{WT/shUGT1}$ cells than HeLaM-TAPBPR$^{C94A}$ cells, perhaps due to additional TAPBPR-independent effects of UGT1 depletion (*Figure 5F*). Together, these results suggest that the loss of the association between TAPBPR and UGT1 reduces the quality control exerted on HLA-A68.

## TAPBPR facilitates the reglucosylation of the glycan on MHC class I molecules

As UGT1 has been shown to promote the reglucosylation of the N-glycan on MHC class I molecules (*Wearsch et al., 2011*; *Zhang et al., 2011*), we next attempted to determine whether the TAPBPR:UGT1 complex enhanced the amount of monoglucosylated oligosaccharide on MHC class I molecules in a cellular environment. The approach we chose was a lectin pulldown using exogenous human calreticulin (hCRT), a chaperone that specifically recognises $Glc_1Man_9GlcNAc_2$ glycans (*Spiro et al., 1996*), followed by blotting for the MHC class I heavy chain. In IFN-γ-treated HeLaM$^{KO}$-TAPBPR$^{WT}$ cells, a substantial increase in the amount of MHC class I molecules recognised by GST-hCRT$^{WT}$ was observed compared to IFN-γ-treated HeLaM$^{KO}$ cells (*Figure 6A*). In IFN-γ-treated HeLaM$^{KO}$TAPBPR$^{C94A}$ cells, the amount of MHC class I molecules detected by GST-hCRT$^{WT}$ was not increased and was comparable to that of HeLaM$^{KO}$ cells (*Figure 6A*). To ensure glycan specificity in these associations, pulldowns with GST-hCRT$^{WT}$ were compared with GST-hCRT$^{Y92A}$, a mutation that disrupts glycan binding (*Thomson and Williams, 2005*; *Del Cid et al., 2010*; *Wearsch et al., 2011*). No association of GST-hCRT$^{Y92A}$ with MHC class I was observed in any of the cell lysates, confirming the glycan specificity of the interactions observed with GST-hCRT$^{WT}$ (*Figure 6A*). These results suggest that the UGT1 bound to TAPBPR$^{WT}$ is capable of reglucosylating the N-linked glycan on MHC class I molecules, regenerating the $Glc_1Man_9GlcNAc_2$ moiety, which can consequently be recognised by calreticulin.

Upon sodium dodecyl sulphate-polyacrylamide gel electrophoresis (SDS-PAGE), the MHC class I heavy chains (HC) in HeLaM cells migrate at different molecular sizes; the upper HC band contains the HLA-B HC (HLA-B15), whereas the lower HC band consists of the HLA-A HC (HLA-A68)

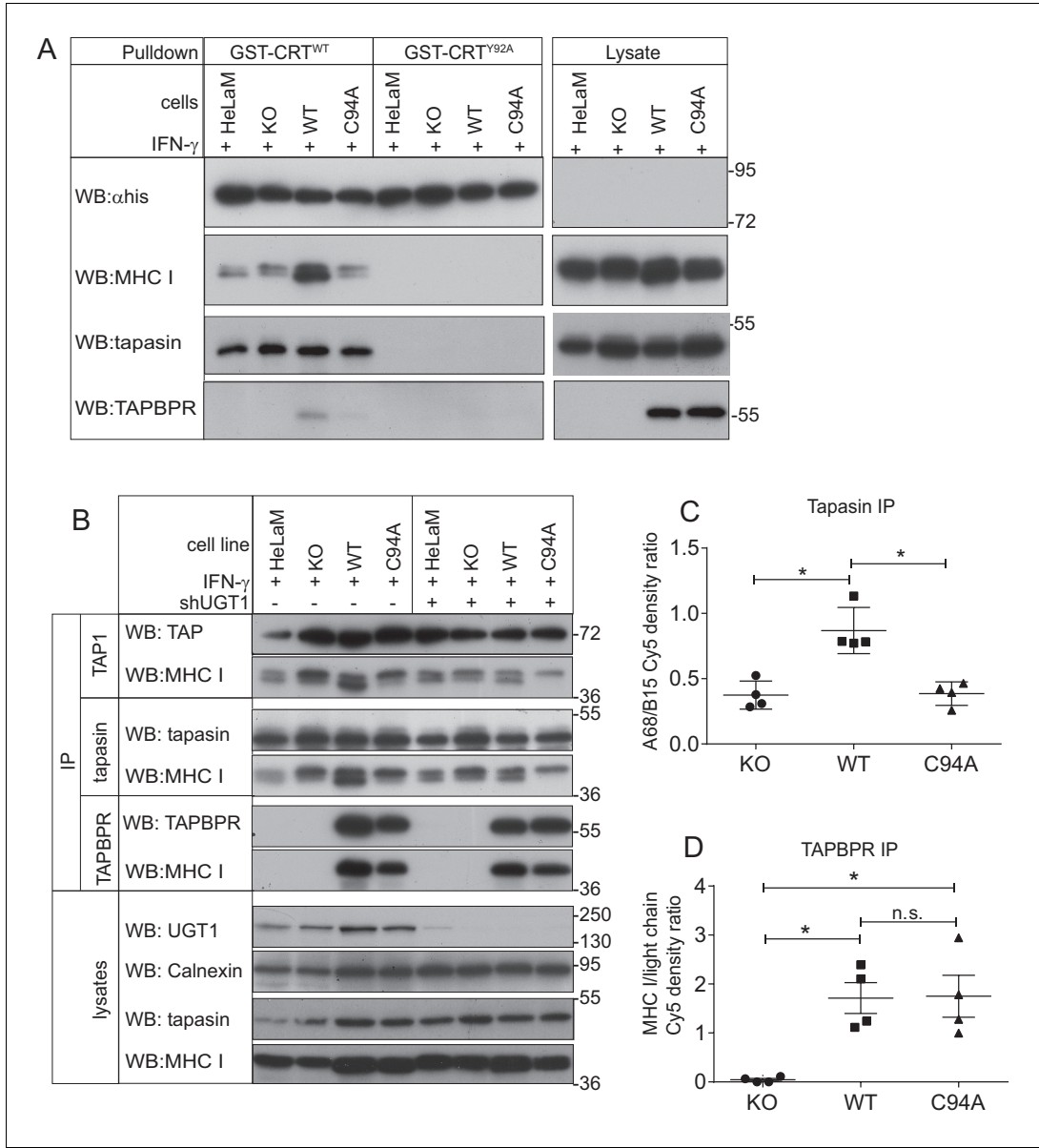

**Figure 6.** The TAPBPR:UDP-glucose:glycoprotein glucosyltransferase 1 complex promotes reglucosylation of MHC class I molecules, enhancing their association with the peptide-loading complex (**A**) Lysates were prepared from IFN-γ-treated HeLaM, HeLaM$^{KO}$, HeLaM$^{KO}$TAPBPR$^{WT}$, and HeLaM$^{KO}$TAPBPR$^{C94A}$ cells in 1% digitonin. After preclear, pulldowns were performed with GST/6xHis-tagged exogenous WT human calreticulin (GST-CRT$^{WT}$), which specifically recognises Glc$_1$Man$_9$GlcNAc$_2$ glycans, or with a CRT variant in which a tyrosine at position 92 had been mutated to alanine (GST-CRT$^{Y92A}$), which disrupts glycan recognition. Western blot analysis was performed for the 6xHis tag, MHC class I HC, tapasin, and TAPBPR on pulldowns and lysates as indicated (see *Figure 6—figure supplement 1* for densitometry analysis on these blots and the Endo H-sensitivity status on the GST-CRT-reactive MHC class I molecules). (**B**) TAP, tapasin, or TAPBPR were isolated by immunoprecipitation from IFN-γ-treated HeLaM, HeLaM$^{KO}$, HeLaM$^{KO}$TAPBPR$^{WT}$, and HeLaM$^{KO}$TAPBPR$^{C94A}$ cells with or without depletion of UGT1 using shRNA. Western blot analysis was performed for TAP, tapasin, TAPBPR, MHC class I HC, UGT1, or calnexin as a loading control on immunoprecipitates or lysates as indicated, resolved under reducing conditions. (**C** and **D**) Quantitative analysis of the MHC class I molecules bound to tapasin and TAPBPR in IFN-γ-treated HeLaM$^{KO}$, HeLaM$^{KO}$TAPBPR$^{WT}$, and HeLaM$^{KO}$TAPBPR$^{C94A}$ cells from four independent Cy5 experiments using the Amersham WB system (see *Figure 6—figure supplement 2* for gel images and further analysis). Scatter dot plots show (**C**) the ratio of HLA-A68 to HLA-B15 associated with tapasin and (**B**) the total amount of MHC class I HC bound to TAPBPR as a ratio of the PeTe4 antibody light chain used in the immunoprecitation. *p<0.05, n.s.

*Figure 6 continued on next page*

*Figure 6 continued*

=not significant based on Mann–Whitney non-parametric, two-tailed tests. KO: knockout; UGT1: UDP-glucose: glycoprotein glucosyltransferase 1; WT: wild-type; IP: immunoprecipitation; WB: western blot.
The following figure supplements are available for figure 6:

**Figure supplement 1.** MHC class I molecules associated with GST-CRT
**Figure supplement 2.** – Densitometry on the MHC class I molecules bound to tapasin and TAPBPR

(*Boyle et al., 2013*). Intriguingly, we observed a difference in the regeneration of $Glc_1Man_9GlcNAc_2$ on the HLA-A and -B molecules by the TAPBPR:UGT1 complex, finding that glucosylation of the N-linked glycan was preferentially increased on the lower MHC class I HC band (i.e. HLA-A*68:02) over the upper MHC class I HC band (i.e. HLA-B*15:03) in cells expressing TAPBPR$^{WT}$ (*Figure 6A*). This is in keeping with previous findings that TAPBPR exhibits prolonged association and a higher affinity for HLA-A molecules (*Boyle et al., 2013*; *Hermann et al., 2015*; *Morozov et al., 2016*). To explore the role of endogenously expressed TAPBPR on the generation of $Glc_1Man_9GlcNAc_2$ on MHC class I molecules, we performed GST-hCRT pulldowns on IFN-γ-treated HeLaM and HeLaM$^{KO}$ cells. In IFN-γ-induced HeLaM cells, the HLA-A68 HC band made up the majority of the MHC class I molecules recognised by calreticulin (*Figure 6A*). In contrast, in IFN-γ-induced HeLaM$^{KO}$ cells, the amount of HLA-A68 recognised by calreticulin was reduced, and proportionally more HLA-B15 had the $Glc_1Man_9GlcNAc_2$ moiety attached (*Figure 6A*). These findings confirm the role of endogenous TAPBPR in modifying the glycan on MHC class I molecules.

## The TAPBPR:UGT1 complex promotes the association of MHC class I with the PLC

As calreticulin escorts MHC class I molecules to the PLC (*Sadasivan et al., 1996*; *Del Cid et al., 2010*; *Wearsch et al., 2011*), next we determined whether the TAPBPR:UGT1 complex enhanced the association of MHC class I molecules with tapasin and TAP. In HeLaM$^{KO}$ cells reconstituted with TAPBPR$^{WT}$, we observed an increase in the amount of MHC class I molecules bound to both tapasin and TAP as compared to HeLaM$^{KO}$cells or cells reconstituted with TAPBPR$^{C94A}$ (*Figure 6B*). In accordance with the results observed above with the calreticulin pulldowns, we specifically observed enhanced PLC association with the lower MHC class I heavy chain band (i.e. HLA-A*68:02) in HeLaM-$^{KO}$TAPBPR$^{WT}$ cells (*Figure 6B*). The increased association of HLA-A68 with tapasin in the presence of TAPBPR$^{WT}$ compared to TAPBPR$^{C94A}$ was independently verified via quantitative analysis (*Figure 6C*), which also confirmed that the amounts of MHC class I molecules bound to TAPBPR$^{WT}$ and TAPBPR$^{C94A}$ were similar (*Figure 6D*). To ensure that the enhanced association of MHC class I with the PLC was specifically due to the association of TAPBPR$^{WT}$ with UGT1, we also determined the association of MHC class I molecules with TAP and tapasin in cells depleted of UGT1 using shRNA. In HeLaM$^{KO}$TAPBPR$^{WT}$ cells depleted of UGT1, the amount of HLA-A68 bound to tapasin and TAP was significantly reduced compared to its UGT1-competent counterpart HeLaM$^{KO}$-TAPBPR$^{WT}$, and was now comparable to HeLaM$^{KO}$TAPBPR$^{C94A}$ without UGT1 depletion (*Figure 6B*). These results suggest that in addition to functioning as an MHC class I peptide editor in its own right, TAPBPR can also influence peptide selection on MHC class I molecules by promoting their association with the PLC, an effect that is dependent on its interaction with UGT1.

## Discussion

Recently, we and others have shown that, like tapasin, TAPBPR can function as a peptide exchange catalyst on MHC class I molecules in vitro (*Hermann et al., 2015*; *Morozov et al., 2016*). However, within a cellular environment, the two related MHC class I chaperones do not appear to be functional equivalents in regard to their influence on peptide selection. While tapasin influences both the loading and optimisation of cargo on MHC class I molecules, TAPBPR appears to have a more subtle, fine-tuning effect on the final peptide repertoire displayed, at least in the limited number of cell lines tested to date. We wondered whether any other co-factors associate with TAPBPR, which

would explain its refining effect on peptide selection in a cellular environment. Here, we identify a novel association between TAPBPR and UGT1, an ER/cis-Golgi-resident enzyme that recognises hydrophobic patches near $Man_9GlcNAc_2$ moieties in incompletely folded or unassembled glycoproteins and transfers a glucose residue from UDP-glucose in order to regenerate $Glc_1Man_9GlcNAc_2$, restoring recognition by the calnexin/calreticulin pathway (*Trombetta et al., 1989*; *Hammond et al., 1994*; *Ware et al., 1995*; *Arnold et al., 2000*; *Tessier et al., 2000*; *Zuber et al., 2001*; *Caramelo et al., 2003*, *Caramelo et al., 2004*; *Ritter et al., 2005*; *D'Alessio et al., 2010*). Human TAPBPR has no N-linked glycan and therefore cannot be reglucosylated by UGT1. Thus, it is highly unlikely that UGT1 functions directly on human TAPBPR. In some species, TAPBPR may contain an N-linked glycan due to the presence of an NxS or NxT motif. However, these are generally found in the IgV domain and therefore might not affect the association with or be effected by UGT1.

In 2011, Cresswell and colleagues first identified a role for UGT1 in the MHC class I pathway (*Wearsch et al., 2011*; *Zhang et al., 2011*). UGT1 was shown to reglucosylate MHC class I molecules associated with suboptimal peptide, thereby permitting re-engagement with the PLC (*Wearsch et al., 2011*). Furthermore, in UGT1-deficient mouse cells, maturation of MHC class I was delayed, surface expression of MHC class I molecules was reduced, and the peptide repertoire presented on MHC class I was impaired (*Zhang et al., 2011*). These studies demonstrate that UGT1 provides a unique level of quality control in the MHC class I presentation pathway, acting as a sensor for the quality of the MHC class I-associated peptide cargo (*Wearsch et al., 2011*; *Zhang et al., 2011*). Our findings here suggest that as well as UGT1 recognising a population of MHC class I molecules in a TAPBPR-independent manner, it can also recognise MHC class I molecules bound to TAPBPR. In this scenario, TAPBPR most likely acts as a platform or bridge between UGT1 and MHC class I. As the MHC class I molecules bound to TAPBPR are in a peptide-receptive state (*Hermann et al., 2015*; *Morozov et al., 2016*), we propose that TAPBPR permits UGT1 to reglucosylate empty or suboptimally loaded MHC class I molecules, thus providing an important quality control checkpoint in the antigen presentation pathway. It is possible that UGT1 senses different conformational perturbations in MHC class I molecules when functioning in a TAPBPR-dependent versus a TAPBPR-independent manner.

Precisely how C94 in TAPBPR permits its binding to UGT1 remains unclear at present. Our evidence does not support a disulphide-mediated interaction between the two proteins. However, perhaps within the environment of the ER and cis-Golgi, a covalent association does indeed exist, but is susceptible in the conditions used to isolate this complex, as was initially observed for the disulphide-mediated bond between tapasin and ERp57, which was found to be extremely labile in the absence of pre-treatment with N-ethymaleimide (NEM) (*Hughes and Cresswell, 1998*; *Dick et al., 2002*). Alternatively, the association between TAPBPR and UGT1 may be dependent on a post-translational modification of C94, such as S-nitrosylation (*Hess et al., 2005*). We have attempted to determine whether C94 is post-translationally modified via mass spectrometry, but we were unable to identify peptides covering C94 when digested with trypsin or chymotrypsin. Interestingly, we found additional residues predicted to form an α-helix on the surface of TAPBPR also influenced its association with UGT1 (*Figure 3—figure supplement 2*), suggesting that C94 may be part of an extended interface. In this capacity, C94 may help stabilise the α-helix as a UGT1 binding site.

We found that the TAPBPR:UGT1 complex promotes the glucosylation of the glycan on MHC class I, regenerating the $Glc_1Man_9GlcNAc_2$ moiety in order to restore recognition by calreticulin, which subsequently results in enhanced association of the MHC class I molecules with the PLC. Therefore, in addition to functioning as a peptide editor in its own right (*Hermann et al., 2015*; *Morozov et al., 2016*), it is now apparent that TAPBPR can also influence peptide selection by promoting peptide-receptive MHC class I molecules to associate with the PLC. Our discoveries here shed light on our previously unexplained observations, namely that, in HeLa cells, fewer HLA-A68 molecules were associated with TAP in TAPBPR-depleted cells compared to their TAPBPR-competent counterparts (*Boyle et al., 2013*). In this previous work, we suggested that the HLA-A68 molecules dissociated faster from the PLC in the absence of TAPBPR, but it now appears that this phenotype is more likely to be explained by a lack of recycling of the HLA-A68 molecules back to the PLC in the absence of TAPBPR due to the lack of reglucosylation of the MHC class I glycan. Our findings here also help to further explain why TAPBPR expression slows down the ER export of MHC class I molecules and enhances their association with tapasin; findings that earlier led us to speculate

that TAPBPR was a second quality control checkpoint in the MHC class I antigen presentation pathway (*Boyle et al., 2013*; *Hermann et al., 2013*).

Although UGT1 is not required for TAPBPR to function as an MHC class I peptide editor, the TAPBPR:UGT1 complex is clearly important for MHC class I peptide optimisation. This is evidenced by the decrease observed in canonical anchor residue prevalence for peptides associated with HLA-A68 in cells expressing TAPBPR$^{C94A}$, a mutant that retains peptide-editing function, but cannot bind UGT1. These results demonstrate that TAPBPR and UGT1 work together to provide a novel quality control checkpoint in the antigen presentation pathway that influences the final peptide repertoire displayed. We propose the following working model (*Figure 7*): in the PLC, MHC class I molecules are loaded with peptides of relatively optimal affinity via tapasin (step 1). However, some MHC class I molecules can also self-assemble with peptides in a tapasin-independent manner, resulting in complexes associated with cargoes of variable quality (step 2). Following glucosidase II-mediated removal of the terminal glucose on the glycan (step 3), some MHC class I molecules can be recognised and reglucosylated directly by UGT1 in a TAPBPR-independent manner (step 4). However, in order to proceed to the cell surface, MHC class I molecules need to pass through the TAPBPR-mediated quality control checkpoint (step 5). MHC class I molecules loaded with high-affinity peptides transit through quickly. However, MHC class I molecules associated with cargoes of low or medium affinity are captured by TAPBPR for a subsequent editing step that favours peptide dissociation. The resultant MHC class I molecule that is devoid of peptide is now reglucosylated by UGT1 associated with TAPBPR (step 6); a modification that sends the MHC class I molecule back into the calnexin/calreticulin pathway, and consequently back to the PLC (step 1) for another attempt at peptide loading. While the reason for TAPBPR preferentially mediating reglucosylation of HLA-A68 over HLA-B15 in HeLaM cells is currently unexplained, it is tempting to speculate that TAPBPR may favour binding to molecules that are initially associated with peptide in a tapasin-independent manner, as has been observed for many HLA-A allomorphs (*Greenwood et al., 1994*), and may be due to the lower affinities of their cargoes.

To conclude, a potential analogous function for unpaired cysteine residues interacting with crucial association partners has been investigated in TAPBPR and tapasin. For tapasin, the free cysteine residue at position C95 is essential for its association with ERp57 in mammals (*Dick et al., 2002*; *Dong et al., 2009*). Tapasin$^{C95A}$ molecules exhibit an impaired ability to load and edit peptides, resulting in poorly loaded MHC class I molecules that escape to the cell surface, where they are unstable (*Dick et al., 2002*). For TAPBPR, its free cysteine residue at position C94 is important for its association with UGT1. TAPBPR$^{C94A}$ molecules exhibit an impaired ability to reglucosylate MHC class I molecules and exhibit a reduction in the ability to optimise peptide selection due to a failure to recycle MHC class I molecules back to the PLC. Thus, the unpaired cysteine residues in both tapasin and TAPBPR in mammals have evolved to perform roles that prevent the release of MHC class I molecules loaded with unsuitable peptides. As a result of functioning as peptide exchange catalysts within distinct cellular environments and their unique association partners, it is becoming evident that tapasin and TAPBPR have evolved to uniquely shape the peptides bound to MHC class I molecules in distinct yet intertwined ways that ultimately determine the final peptide repertoire presented on the surface of cells.

## Materials and methods

### Constructs

The production of full-length TAPBPR$^{WT}$ and TAPBPR$^{TN5}$ in the lentiviral vector pHRSIN-C56W-UbEM, which produces TAPBPR under the control of the spleen focus-forming virus (SFFV) promoter and the GFP derivative emerald under the control of a ubiquitin promoter, has been previously described (*Boyle et al., 2013*; *Hermann et al., 2013*). To mutate individual cysteine residues in TAPBPR, site-directed mutagenesis was performed on untagged TAPBPR in pCR-Blunt II-TOPO (Thermo Fisher Scientific, UK) using Quik-Change site-directed mutagenesis (Stratagene, La Jolla, California) together with the primers specified in *Table 3*. All TAPBPR variants were subsequently cloned into pHRSIN-C56W-UbEM. A variant of pHRSIN-C56W-UbEM containing a generic N-terminal leader sequence, two protein A cassettes (ZZ), a tobacco etch virus (TEV) protease cleavage site, and a myc tag followed by an insertion site was produced. cDNA encoding amino acids 22–468 of

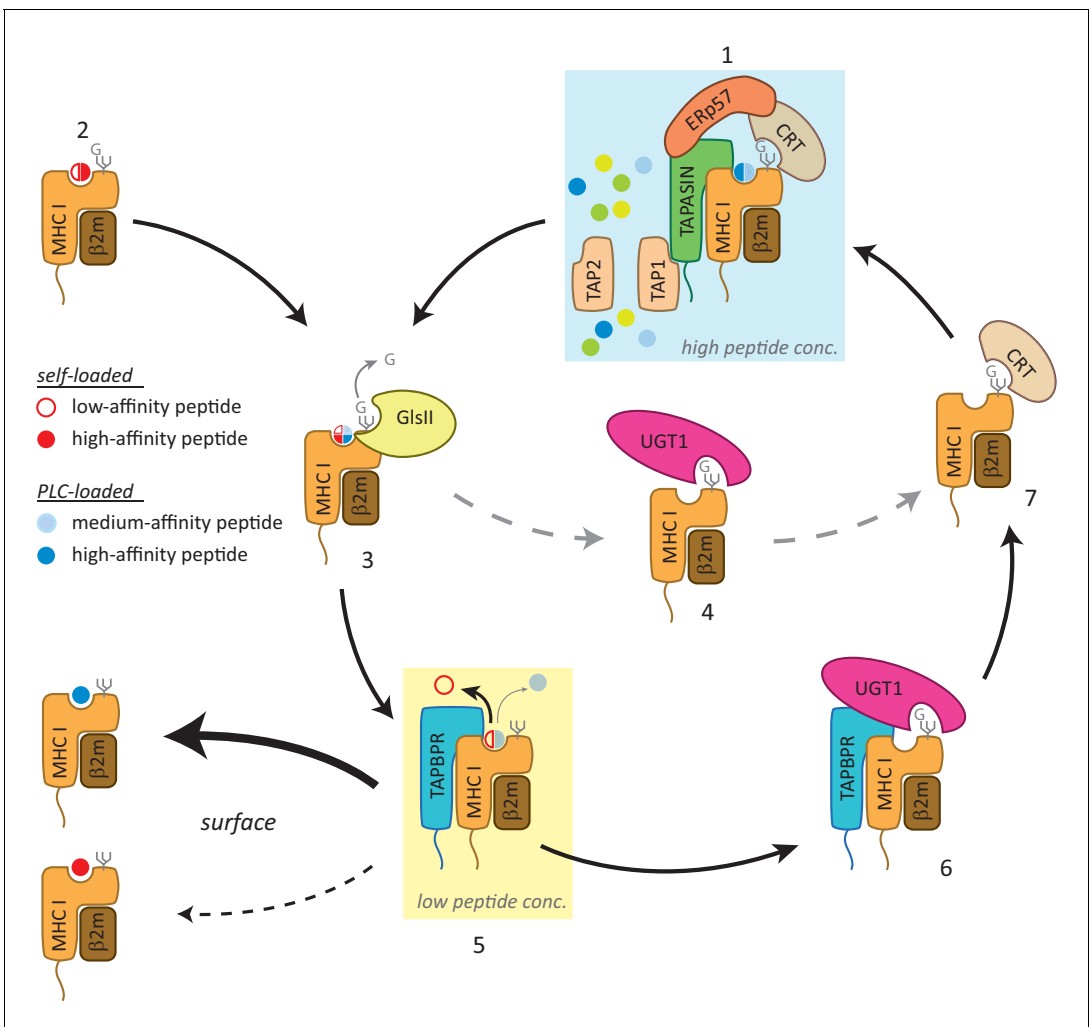

**Figure 7.** Working model of the TAPBPR:UDP-glucose:glycoprotein glucosyltransferase 1 complex in the MHC class I presentation pathway. In addition to (1) the loading and editing of peptides (shown in blue) via tapasin in the peptide-rich milieu of the peptide-loading complex (PLC), an environment that favours MHC class I molecules associating with a broad range of optimal peptides, (2) some MHC class I molecules can also self-assemble with peptides with a wide range of affinities (shown in red) in the endoplasmic reticulum (ER) in a tapasin-independent manner. (3) Regardless of the folding state or affinity/source of the cargo occupying the peptide binding groove (reflected here as a quartered circle to signify a broad range of peptide affinity from various sources of loading), GlsII will mediate removal of the terminal glucose on the glycan in the absence of protection by calnexin/calreticulin. (4) Some conformations of MHC class I molecules will be recognised and reglucosylated by UGT1 in a TAPBPR-independent manner. (5) In order to be exported to the cell surface, MHC class I molecules need to pass through the TAPBPR-mediated quality control checkpoint. If a high-affinity peptide is bound to the MHC class I molecule, TAPBPR either does not bind to the MHC class I molecule or associates transiently and is quickly released, permitting the MHC class I molecule to proceed through the secretory pathway. If the MHC class I molecule is associated with a cargo of low or medium affinity, TAPBPR-mediated peptide editing occurs. In contrast to tapasin-mediated peptide editing in the PLC, this second TAPBPR-mediated editing step favours peptide dissociation either as a consequence of occurring in more peptide-restrictive areas of the ER, cis-Golgi, and medial Golgi or due to a distinct functionality of TAPBPR. (6) MHC class I molecules devoid of peptide are now recognised and reglucosylated by UGT1 associated with TAPBPR. (7) This modification on the N-linked glycan sends the MHC class I molecule back into the calnexin/calreticulin pathway, and consequently back to the PLC (1) for another attempt at peptide loading. In this scenario, TAPBPR may exhibit preferential binding to MHC class I molecules that associate with peptides in a tapasin-independent manner, as has been observed for many HLA-A allomorphs. GlsII: glucosidase II; UGT1: UDP-glucose:glycoprotein glucosyltransferase 1.

human TAPBPR (i.e. TAPBPR minus its own signal sequence) was cloned into this vector downstream of a GAGA linker sequence. This produces a protein product termed ZZ-TAPBPR, in which the mature TAPBPR protein is tagged at the N-terminus with two protein A tags (*Boyle et al., 2013*).

**Table 3.** Primer sequences used for the mutation of individual cysteine residues to alanine in TAPBPR

| Name | Primers used for site-directed mutagenesis | Predicted TAPBPR domain |
|---|---|---|
| C18A | 5'-CAGTGGACGTGGTCCTAGACGCTTTCCTGGTGAAGGACGGTG-3'<br>5'-CACCGTCCTTCACCAGGAAAGCGTCTAGGACCACGTCCACGT-3' | Unique N-terminal |
| C94A | 5'-GAGGCCTTGCTCCATGCTGACGCCAGTGGGAAGGAGGTGACCTG-3'<br>5'-CAGGTCACCTCCTTCCCACTGGCGTCAGCATGGAGCAAGGCCTC-3' | |
| C101A | 5'-CTGCAGTGGGAAGGAGGTGACCGCTGAGATCTCCCGCTACTTTCTC-3'<br>5'-GAGAAAGATGCGGGAGATCTCAGCGGTCACCTCCTTCCCACTGCAG-3' | |
| C191A | 5'-GGTCCTCAGCCTCCTTGGACGCTGGCTTCTCCATGGCACCGG-3'<br>5'-CCGGTGCCATGGAGAAGCCAGCGTCCAAGGAGGCTGAGGACC-3' | IgV domain |
| C262A | 5'-CAGGACGAGGGGACCTACATTGCCCAGATCACCACCTCTCTGTAC-3'<br>5'-GTACAGAGAGGTGGTGATCTGGGCAATGTAGGTCCCCTCGTCCTG-3' | |
| C300A | 5'-GCTCTGCTGCCCACCCTCATCGCCGACATTGCTGGCTATTACC-3'<br>5'-GGTAATAGCCAGCAATGTCGGCGATGAGGGTGGGCAGCAGAGC-3' | IgC domain |
| C361A | 5'-CTGCAGGTGCAACTTACACCGCCCAGGTCACACACATCTCTC-3'<br>5'-GAGAGATGTGTGTGACCTGGGCGGTGTAAGTGGCACCTGCAG-3' | |
| C427A | 5'-GAACGCTGGGAGACCACTTCCGCTGCTGACACACAGAGCTCCC-3'<br>5'-GGGAGCTCTGTGTGTCAGCAGCGGAAGTGGTCTCCCAGCGTTC-3' | Cytoplasmic tail |

The luminal domains of TAPBPR[C94A] were also cloned into pHLsec to produce a secreted form of this TAPBPR variant containing a C-terminal His tag in a mammalian expression system (*Aricescu et al., 2006*). For shRNA-mediated depletion of TAPBPR, the V2LHS_135531 plasmid (GE Healthcare, UK) was used (mature antisense sequence: ATTCCTACCATTAAACTGG). For shRNA-mediated knockdown of human UGT1 expression, hairpin oligonucleotides shUGT1-1-for (5'- GA TCCGCAGTAAAGGCCGACTCAAATTCAAGAGATTTGAGTCGGCCTTTACTGTTTTTTG-3') and shUGT1-1-rev (5'-AATTCAAAAAACAGTAAAGGCCGACTCAAATCTCTTGAATTTGAGTCGGCC TTTACTGCG-3') were annealed and cloned into the pHR-SIREN/Puro lentiviral vector (a kind gift from Professor Greg Towers, University College London, UK) cut with BamHI and EcoRI, and sequence verified. For knockout of TAPBPR and tapasin, sgRNAs were chosen using the CRISPR design tool (http://crispr.mit.edu), cloned into pSpCas9(BB)−2A-puro and transfected as previously described (*Ran et al., 2013*; *Hermann et al., 2015*). sgRNA-Crispr7 (GCGAAGGACGG TGCGCACCG) and sgRNA-TAPA8 (GGTGCACTGCTGTTGCGCCA) were used to bind to exon 2 of TAPBPR and tapasin, respectively.

## Cell culture

HeLaM cells, a variant HeLa cell line that is more responsive to IFN (*Tiwari et al., 1987*) (a kind gift from Paul Lehner, University of Cambridge, UK), their modified variants, and HEK-293T (from Paul Lehner, University of Cambridge, UK) were maintained in Dulbecco's Modified Eagle's medium (DMEM; Sigma-Aldrich, UK), while the near-haploid human chronic myeloid leukaemia cell line KBM-7 (*Kotecki et al., 1999*) and its variants (*Duncan et al., 2012*) (kind gifts from Lidia Duncan and Paul Lehner, University of Cambridge, UK) were cultured in Iscove's Modified Dulbecco's Medium (IMDM) media (Gibco, Thermo Fisher Scientific, UK), both supplemented with 10% fetal calf serum (Gibco, Thermo Fisher Scientific), 100 U/ml penicillin and 100 μg/ml streptomycin (Gibco, Thermo Fisher Scientific) at 37°C with 5% $CO_2$. All cells were confirmed to be mycoplasma negative (MycoAlert, Lonza, UK). Although cell line authentication using short tandem repeat profiling was not undertaken for this work, the authenticity of HeLaM and KBM-7 cells was verified by the continuous confirmation that these cell lines had the expected HLA class I tissue type monitored by both staining with specific HLA antibodies and by mass spectrometry. To induce/up-regulate TAPBPR expression and other components of the MHC class I antigen processing and presentation pathway, HeLaM and KBM-7 cells were treated with 100 U/ml IFN-γ (Peprotech, UK) where indicated for 48–72 h.

## Lentiviral transductions and transfections

Lentivirus was produced by transfecting HEK-293T cells with lentiviral vectors along with the packaging vector pCMVΔR8.91 and the envelope vector pMD.G using TransIT-293 (Mirus, Madison, WI). Viral supernatant was collected at 48 and 72 h and used to transduce HeLaM cells. shTAPBPR- or shUGT1-depleted cells were selected, and subsequently maintained, in medium containing puromycin. A HeLaM TAPBPR-knockout cell line (HeLaM$^{KO}$) was generated using the CRISPR-Cas9 system as previously described (*Hermann et al., 2015*). TAPBPR$^{WT}$, TAPBPR$^{C94A}$, and TAPBPR$^{TN5}$ were reconstituted in the HeLaM$^{KO}$ cells via transduction.

## Antibodies

The following TAPBPR-specific antibodies were used: PeTe4, a mouse monoclonal antibody (mAb) specific for the native conformation of TAPBPR raised against amino acids 22–406 of human TAPBPR (*Boyle et al., 2013*) that does not cross-react with tapasin (*Hermann et al., 2013*); R021, a rabbit polyclonal raised against the cytoplasmic tail of human TAPBPR (*Hermann et al., 2013*); and ab57411, a mouse mAb raised against amino acids 23–122 of TAPBPR that is reactive to denatured TAPBPR (Abcam, UK).

The following MHC class I-specific antibodies were used: W6/32, a pan-MHC class I mAb that recognises a conformation-specific epitope on the MHC class I α2 domain in a manner that is dependent on β2m and peptide (*Barnstable et al., 1978*); HC10, a MHC class I-specific mAb that recognises HLA-A, -B, and -C molecules containing a PxxWDR motif at amino acids 57–62 in the α1 domain (*Stam et al., 1986*; *Perosa et al., 2003*); 3B10.7, a rat mAb with broad specificity for HLA I independent of conformation (*Lutz and Cresswell, 1987*); anti-HLA-A68-reactive mAb, specific for HLA-A2 and -A68 heavy chain/β2m heterodimers (One Lambda, Thermo Fisher Scientific, Canoga Park, CA); and 4E, a mouse mAb reactive against HLA-B as well as a limited number of HLA-A molecules (A29, Aw30, Aw31, and Aw32) (*Yang et al., 1984*).

Other antibodies used were: Pasta-1, the tapasin-specific mAb (*Dick et al., 2002*); R.gp48N, a rabbit polyclonal antibody to tapasin (*Sadasivan et al., 1996*); rabbit anti-calnexin (Enzo Life Sciences, UK); rabbit mAb to UGT1 (ab124879, Abcam); and Ring4C, a rabbit anti-peptide antibody raised to the C-terminal region of TAP1 (*Ortmann et al., 1994*; *Meyer et al., 1994*). Isotype control antibodies (Dako, UK), horseradish peroxidase (HRP)-conjugated species-specific secondaries (Dako and Rockland Immunochemicals Inc., Limerick, PA), and species-specific Alexa-Fluor secondary antibodies (Invitrogen Molecular Probes Thermo Fisher Scientific) were also used.

## Immunoprecipitation, gel electrophoresis, and western blotting

Cells were harvested then washed in phosphate-buffered saline (PBS) supplemented with 10 mM NEM (Sigma-Aldrich, UK). For TAPBPR and MHC class I immunoprecipitation experiments, cells were lysed in 1% digitonin (Merck Millipore, Germany), Tris-buffered saline (TBS) (20 mM Tris-HCl, 150 mM NaCl, 5 mM MgCl$_2$, 1 mM EDTA) supplemented with 10 mM NEM, 1 mM phenylmethylsulfonyl fluoride (PMSF) (Sigma-Aldrich), and protease inhibitor cocktail (Roche, UK) for 30 min at 4°C. For pulldown experiments using recombinant GST-tagged hCRT$^{WT}$ or hCRT$^{Y92A}$ (both kind gifts from Najla Arshad and Peter Creswell, Yale University School of Medicine, New Haven, CT) (*Wearsch et al., 2011*) or tapasin and TAP immunoprecipitations, cells were lysed in 1% digitonin TBS in which 5 mM MgCl$_2$ and EDTA were omitted and replaced with 2.5 mM CaCl$_2$. Nuclei and cell debris were pelleted by centrifugation at 13,000 × g for 10 min and supernatants were pre-cleared on IgG-sepharose (GE Healthcare) and protein A sepharose (Generon, UK) for 1 h at 4°C with rotation. Immunoprecipitation was performed with the indicated antibody and protein A sepharose or recombinant calreticulin with glutathione sepharose (GE Healthcare) for 2–3 h at 4°C with rotation. Following immunoprecipitation, beads were washed thoroughly in 0.1% detergent-TBS to remove unbound protein. Samples for separation by gel electrophoresis were heated at 94°C for 10 min in sample buffer (125 mM Tris-HCl pH 6.8, 4% SDS, 20% glycerol, 0.04% bromophenol blue). For reducing SDS-PAGE, sample buffer was supplemented with 100 mM β-mercaptoethanol. For samples to be analysed by western blotting, proteins were transferred onto an Immobilon transfer membrane (Merck Millipore). Membranes were blocked using 5% (w/v) dried milk and 0.1% (v/v) Tween 20 in PBS for 30 min, followed by incubation with the indicated primary antibody for 1–16 h. After washing, membranes were incubated with species-specific HRP-conjugated secondary antibodies,

washed, and detected by enhanced chemiluminescence using Western Lightning (Perkin Elmer, UK) and Super RX film (Fujifilm, UK). Films were scanned on a CanoScan8800F using MX Navigator Software (Canon, UK). For comparative analysis, the pixel densities of three independent scans were analysed using ImageJ. The background was determined and subtracted by using two random locations on the scanned film.

## Quantification of MHC class I molecules associated with tapasin and TAPBPR

Tapasin and TAPBPR immunoprecipitates were pre-labelled with Cy5 fluorescent dye (GE Healthcare) and subjected to electrophoresis using the Amersham WB system. Cy5 total protein images from four separate electrophoresis experiments were analysed by ImageJ. Densitometry graphs from relevant tracks were subsequently analysed in MATLAB. For the tapasin immunoprecipitate, a MATLAB script was developed (see *Source code 1*) and applied in order to calculate the densities of the A68 and B15 bands.

## Affinity chromatography using IgG-sepharose

To identify association partners for TAPBPR, affinity chromatography using IgG-sepharose was performed on HeLaM cells transiently transfected with ZZ-TAPBPR-pHRSIN-C56W-UbEM. As a control, HeLaM cells transiently transfected with an empty pHRSIN-C56W-UbEM were used. Cells were lysed at 4°C in 1% digitonin in TBS plus 1 mM PMSF, 10 mM NEM, and protease inhibitor cocktail; nuclei and cell debris were removed by centrifugation at 17,000 × g for 10 min at 4°C, and then supernatants were pre-cleared on sepharose beads (GE Healthcare). ZZ-TAPBPR and associated proteins were immunoprecipitated by incubating with IgG-sepharose beads (GE Healthcare) for 2 h at 4°C, followed by thorough washing in 0.1% digitonin TBS.

## Protein identification by mass spectrometry

Immunoprecipitated samples were run a short distance into a pre-cast 4–12% polyacrylamide gel and each lane was cut into four approximately equally sized slices. Proteins were digested in gel and the tryptic peptides were eluted for analysis using an Orbitrap XL (Thermo) coupled to a NanoAcquity Ultra Performance Liquid Chromatography (Waters, UK). Peptides were eluted using a gradient rising from 8% to 25% acetonitrile (MeCN) for 27 min and 40% MeCN for 35 min. Mass spectra were acquired at 60,000 fwhm at between 300 and 2000 m/z, with MS/MS spectra acquired by top 6 collision-induced dissociations. Raw files were searched against a human Uniprot database (download 041113, 68,896 entries) using the Mascot search engine, with carbamidomethyl cysteine as a fixed modification and acetyl N-terminus and oxidised methionine as variable modifications. Scaffold (version Scaffold_4.3.2, Proteome Software, Inc., Portland, OR) was used to validate MS/MS-based peptide and protein identifications. Peptide identifications were accepted if they could be established at greater than 90.0% probability. Peptide Probabilities from X! Tandem were assigned by the Scaffold Local false discovery rate algorithm. Peptide Probabilities from Mascot were assigned by the Peptide Prophet algorithm (*Keller et al., 2002*) with Scaffold δ-mass correction. Protein identifications were accepted if they could be established at greater than 95.0% probability. Protein probabilities were assigned by the Protein Prophet algorithm (*Nesvizhskii et al., 2003*). Proteins that contained similar peptides and could not be differentiated based on MS/MS analysis alone were grouped in order to satisfy the principles of parsimony.

## Expression and purification of TAPBPR proteins

The luminal domains of TAPBPR$^{WT}$ and TAPBPR$^{C94A}$ were cloned into pHLsec, expressed in HEK293F cells (Invitrogen, Thermo Fisher Scientific, UK), purified using Ni-NTA affinity chromatography (Invitrogen, Thermo Fisher Scientific), and separated by size exclusion chromatography as previously described (*Hermann et al., 2015*). Protein-containing fractions were analysed by SDS-PAGE followed by Coomassie staining, pooled, and further concentrated. The concentrate was snap frozen in liquid nitrogen and stored at −80°C.

## Differential scanning fluorimetry

Differential scanning fluorimetry (DSF) experiments on purified TAPBPR[WT] and TAPBPR[C94A] were performed as previously described (*Hermann et al., 2015*). The protein melting temperature ($T_m$) was taken as the inflexion point of the sigmoidal melting curve, obtained by curve fitting using DSF scripts (*Niesen et al., 2007*).

## MHC class I binding peptides

The following MHC class I-specific peptides were used: FEC peptide pool (repository reference: ARP7099); a panel of 32 high-affinity HLA-A and -B binding peptides derived from influenza, Epstein–Barr virus (EBV), and cytomegalovirus (CMV) viral proteins (*Currier et al., 2002*) (from the National Institute for Biological Standards and Control, Potters Bar, UK); HLA-A2 binding peptide NLVPMVATV (from Peptide Synthetics, UK); UV-labile peptide KILGFVFjV (where j denotes 3-amino-3-[2-nitro] phenyl-propionic acid) (from Peptide Synthetics); and the fluorescent peptide FLPSDC*FPSV (C* denotes 5-carboxytetramethylrhodaime [TAMRA]-labelled cysteine) (from GL Biochem Ltd, Shanghai, China).

## Fluorescence polarization experiments

HLA-A*02:01 or HLA-A*02:01fos heavy chains were refolded with human $\beta$2m and KILGFVFjV as previously described (*Hermann et al., 2015*). To measure the peptide dissociation rate, peptide-receptive HLA-A*02:01 was obtained by exposing 500 nM monomeric HLA-A*02:01 complexes to 366-nm light for 20 min at 4°C ('UV-exposed' hereafter). UV-exposed HLA-A*02:01 molecules were allowed to bind 17.6 nM of the fluorescent peptide FLPSDC*FPSV overnight at room temperature. Dissociation of FLPSDC*FPSV was subsequently followed after the addition of 1000-fold molar excess non-labelled NLVPMVATV competitor peptide in the absence or presence of 0.125 µM TAPBPR[WT] or TAPBPR[C94A]. Fluorescence polarisation measurements were taken using an I3x (Molecular Devices, Sunnyvale, CA) with a rhodamine detection cartridge or an Analyst AD (Molecular Devices). All experiments were conducted at room temperature (using the Analyst AD) or 25°C (using the I3x) and used PBS supplemented with 0.5 mg/ml bovine γ-globulin (Sigma-Aldrich) in a volume of 60 µl. Binding of TAMRA-labelled peptides is reported in millipolarisation units (mP) and is obtained from the equation mP = $1000 \times (S - G \times P) / (S + G \times P)$, where S and P are background subtracted fluorescence count rates (S = polarisation emission filter is parallel to the excitation filter; P = polarisation emission filter is perpendicular to the excitation filter) and G (grating) is an instrument- and assay-dependent factor. To measure the association rate of the peptides, 75 nM HLA-A02:01fos molecules were mixed with 1.5 µM human $\beta$2m and exposed to 366-nm UV light at 4°C for 20 min, and then 5.95 nM FLPSDC*FPSV was added in the absence or presence of 0.125 µM TAPBPR[WT] or TAPBPR[C94A] immediately before fluorescence polarisation measurements were taken. For peptide competition experiments, 75 nM HLA-A02:01fos molecules were mixed with 1.5 µM human $\beta$2m and exposed to 366-nm UV light at 4°C for 20 min, and then incubated with 5.95 nM high-affinity peptide FLPSDC*FPSV and various concentrations of the lower-affinity competing peptide NLVPMVATV (0–100 µM) in the presence or absence of 0.0625 µM TAPBPR[WT] or TAPBPR[C94A]. Fluorescence polarisation measurements were taken after incubation overnight at 25°C.

## Flow cytometry

Following trypsinisation, cells were allowed to recover in complete media at 37°C or 26°C for 90 min supplemented with or without 100 µM FEC peptide pool (resulting in individual peptides present at a final concentration of 3 µg/ml), as indicated. Following washing in 1× PBS at 4°C, cells were stained at 4°C for 25 min in 1% bovine serum albumin/PBS with MHC class I-specific antibodies or with an isotype control antibody. After washing the cells to remove excess unbound antibody, primary antibodies bound to the cells were subsequently detected by incubation at 4°C for 25 min with goat anti-mouse Alexa-Fluor 647 (Invitrogen Molecular Probes, Thermo Fisher Scientific). Fluorescence was detected after washing using a BD FACScan analyser with Cytek modifications and analysed using FlowJo (FlowJo, LLC, Ashland, OR).

## MHC class I peptide analysis

HLA ligands from $5 \times 10^8$ IFN-γ-treated HeLaM$^{KO}$, HeLaM$^{KO}$TAPBPR$^{WT}$, and HeLa$^{KO}$TAPBPR$^{C94A}$ cells were isolated by immunoaffinity chromatography using W6/32, eluted, and analysed by MS/MS analysis in a LTQ Orbitrap XL instrument (ThermoFisher Scientific, Bremen, Germany) as previously described (*Hermann et al., 2015*).

## Graphs and statistical analysis

GraphPad Prism (GraphPad Software, La Jolla, CA, www.graphpad.com) was used to generate graphs and for statistical analysis.

## Acknowledgements

We are extremely grateful to Peter Cresswell and Najla Arshad (Yale University School of Medicine, New Haven, CT) for valuable advice, tapasin and TAP-specific antibody reagents, and the recombinant calreticulin proteins. We thank John Trowsdale (University of Cambridge, UK) for his mentorship and critical reading of this manuscript, and Jim Kaufman (University of Cambridge, UK) for useful discussions. We also thank Yi Cao (Cranfield University, UK) for MATLAB programming for densitometry analysis, and Mark Vickers and Sadie Henderson (Scottish National Blood Transfusion Services, UK) for permitting the use of and assistance with the Amersham WB system. The reagent ARP7099 FEC peptide pool was obtained from the Centre for AIDS Reagents, National Institute for Biological Standards and Control (NIBSC), and was donated by the NIH AIDS Reagent Program, Division of AIDS, NIAID, NIH.

## Additional information

### Funding

| Funder | Grant reference number | Author |
|---|---|---|
| Wellcome | Senior Research Fellowship 104647 | Andreas Neerincx Louise H Boyle |
| Royal Society | University Research Fellowship,UF100371 | Janet E Deane |
| Cancer Research UK | Programme Grant,C7056A | Andy van Hateren Tim Elliott |
| Deutsche Forschungsgemeinschaft | SFB 685 | Nico Trautwein Stefan Stevanović |
| Wellcome | PhD studentship,089563 | Clemens Hermann |
| Wellcome | Strategic Award 100140 | Robin Antrobus |
| Wellcome | programme grant, WT094847MA | Huan Cao |

The funders had no role in study design, data collection and interpretation, or the decision to submit the work for publication.

### Author contributions

AN, Data curation, Formal analysis, Validation, Investigation, Visualization, Writing—review and editing; CH, Conceptualization, Data curation, Formal analysis, Validation, Investigation, Writing—review and editing; RA, NT, Data curation, Formal analysis; AvH, Data curation, Formal analysis, Validation, Investigation, Writing—review and editing; HC, Data curation, Formal analysis, Investigation; SS, Supervision, Funding acquisition; TE, Supervision, Funding acquisition, Writing—review and editing; JED, Data curation, Formal analysis, Supervision, Writing—review and editing; LHB, Conceptualization, Data curation, Formal analysis, Supervision, Funding acquisition, Validation, Investigation, Writing—original draft, Project administration, Writing—review and editing

## Author ORCIDs

Andy van Hateren, http://orcid.org/0000-0002-3915-0239
Janet E Deane, http://orcid.org/0000-0002-4863-0330
Louise H Boyle, http://orcid.org/0000-0002-3105-6555

## Additional files

### Supplementary files

• Supplementary file 1. Primers used to generated TAPBPR[UBS1]- and TAPBPR[UBS2]-variant molecules
To create the UDP-glucose:glycoprotein glucosyltransferase 1 binding site mutants, site-directed mutagenesis was performed on untagged TAPBPR in pCR-Blunt II-TOPO using Quik-Change site-directed mutagenesis (Stratagene) together with the primers specified in this table. The resultant TAPBPR[UBS1] (I83K and E87K) and TAPBPR[UBS2] (E87K, L90K, H91S, and D93R) variants were subsequently cloned into pHRSIN-C56W-UbEM and transduced into HeLaM cells.

• Source code 1. A MATLAB script was developed and applied in order to calculate the densities of the HLA-A68 and -B15 bands bound to tapasin in Pasta1 immunoprecipitation experiments. The maximum peak corresponding to B15 was aligned relative to that of the wild-type (WT) track because the separation of B15 and A68 is most distinctive. The separation point between B15 and A68 was identified in the WT track as the minimum between the two peaks. This distance between the B15 peak and the B15/A68 separation point as found in WT was calculated and applied to the alignment positions on the knockout and C94A tracks in order to separate B15 and A68 in these two tracks. Areas under the respective curves generated the densities of the corresponding MHC class I molecules.

### Major datasets

The following dataset was generated:

| Author(s) | Year | Dataset title | Dataset URL | Database, license, and accessibility information |
|---|---|---|---|---|
| Neerincx A, Hermann C, Antrobus R, van Hateren A, Huan C, Trautwein N, Stevanović S, Elliott T, Deane JE, Boyle LH | 2016 | Data from: TAPBPR bridges UDP-glucose:glycoprotein glucosyltransferase 1 onto MHC class I to provide quality control | http://dx.doi.org/10.5061/dryad.8fh3g | Available at Dryad Digital Repository under a CC0 Public Domain Dedication |

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
