## [Decision Letter]

Thank you for submitting your article "TAPBPR bridges UDP-glucose:glycoprotein glucosyltransferase 1 onto MHC class I to provide quality control" for consideration by *eLife*. Your article has been favorably evaluated by Tadatsugu Taniguchi (Senior Editor) and three reviewers, one of whom is a member of our Board of Reviewing Editors. The following individuals involved in review of your submission have agreed to reveal their identity: Pamela Wearsch (Reviewer #2); Frank Homburg (Reviewer #3).

The reviewers have discussed the reviews with one another and the Reviewing Editor has drafted this decision to help you prepare a revised submission.

This paper describes the association of tapasin binding protein, related (TAPBPR), a currently characterized peptide editor/chaperone, with UDP-glucose: glycoprotein glycosyltransferase (UGT1), an enzyme involved in quality control of partially folded molecules in the secretory pathway. The relevance to antigen presentation via the MHC-I pathway is high, and the current paper adds important information to the development of mechanistic models for the role of TAPBPR in the context of the classical tapasin-containing peptide loading complex (PLC). The role of an unpaired cysteine 94 in TAPBPR is established although the mechanism of its contribution remains unclear. This result forms the basis of an evolving model for the function of TAPBPR as it contributes to post-endoplasmic reticulum editing for high affinity peptides. The work identifies a novel molecular interaction, documents this extensively with biochemical and cell biological evidence, and uses this to develop a sensible mechanistic model. The observations are established by pull-down experiments in cells and evaluation of knock down and overexpressing cell lines, and uses recombinant proteins to further substantiate the observations.

Nevertheless, a number of shortcomings were noted, ranging from minor experimental details, to lack of adequate statistical analysis, to aspects that may require additional experimentation, to concerns about the basis and the validity of the final model.

Note that Reviewer #2 offers considerable criticism of the model, and appropriate rebuttal or modification of the model should be entertained.

*Reviewer #1:*

The work is done well, taking advantage of unique reagents, but my major concern is that a mechanistic explanation for the role of C94 is not clear, and is not sufficiently addressed in Discussion. In addition, a number of nagging details in the description of the reagents, in the referencing, and in experimental controls should be addressed. One issue not dealt with in the Discussion or the model is the quantitative relationships of tapasin/MHC interaction as compared with TAPBPR interaction and the relative levels of expression of tapasin and TAPBPR in different compartments and different tissues.

*Reviewer #2:*

This manuscript reports a novel interaction between TAPBPR and UGT1 and provides mechanistic explanations for previously unexplained phenomena regarding the function of TAPBPR in antigen processing. Overall, the experiments are well-designed and executed, and the manuscript is well-written. Although I generally agree with the interpretation of the data in the Results section, I have significant questions and concerns about the model (Figure 6) that is presented in the Discussion section. I acknowledge the challenge of incorporating decades of research into a model, but believe that we can generate one that is more simplistic and comprehensive of past and present data.

Issues regarding the model:

The difference in tapasin-dependence for HLA-A vs HLA-B alleles is a highly relevant issue, but largely ignored. HLA-A molecules are not tapasin-dependent (Greenwood 1994) and do not necessarily interact with the PLC prior to deglucosylation by GLSII (step 1). Self-loaded MHC molecules are therefore an alternate entry point into the "TAPBPR cycle". This is supported by the finding that the TAPBPR-MHC interaction is increased in the absence of tapasin (Boyle 2013). Furthermore, self-loaded HLA-A molecules are more likely to be TAPBPR substrates than PLC-loaded HLA-A molecules.

The major flaw (and constraint) of this model is the authors' view that tapasin and TAPBPR are functionally equivalent in regards to peptide editing.

A) If true, what would determine whether TAPBPR association with HLA-A leads to step 4 (peptide exchange) vs. step 5 (peptide displacement and UGT1 association)? If true, why would a redundancy of tapasin/TAPBPR function be energetically favorable? In other words, why go through additional steps of UGT1 reglucosylation and PLC association (steps 5 and 6) if TAPBPR can do the same thing all by itself (step 4)?

B) Keeping in mind the number of studies on tapasin vs. TAPBPR, I think it's premature to conclude that their functional activities are identical. Although I would agree that TAPBPR stabilizes HLA-A molecules and promotes the displacement of peptides, the "selectivity" of TAPBPR activity has yet to be convincingly demonstrated (Hermann 2015, Morozov 2016, this study). More comprehensive studies are needed to directly compare tapasin and TAPBPR activities using a broader panel of low and high affinity peptides.

C) Another oversight of this model (and past studies of TAPBPR) is the peptide supply. Immunofluorescence and EndoH experiments (Boyle 2013) show that TABPBR is not exclusively localized to the ER. Even within the ER, the peptide supply may be significantly higher near TAP. Thus, the localization of TAPBPR vs. tapasin may have a profound effect on their potential to edit the peptide repertoire in vivo.

Revised model:

I propose an alternative model for discussion with the authors and reviewers.

1) Some HLA-A molecules assemble with peptides independently of the PLC. Of these, some bind high affinity peptides and some bind suboptimal peptides.

2) HLA-A molecules that have bypassed the PLC are more rapidly trimmed by GlsII because the glycan is not "protected" by calreticulin. Self-loading may have occurred before or after GlsII trimming. (I do not recall evidence that would support one option vs. the other).

3) HLA-A molecules loaded with high affinity peptides pass quality control and do not interact with TAPBPR.

4) HLA-A molecules loaded with suboptimal peptides associate with TAPBPR in the ER or cis-Golgi.

5) Due to differences in TAPBPR vs. tapasin activity and/or the peptide supply, TAPBPR promotes the dissociation of the suboptimal ligand, but does not load a high affinity peptide.

6) Instead, TAPBPR improves the efficiency of reglucosylation by recruiting UGT1. Another advantage of the TAPBPR interaction is that HLA-A molecules are stabilized before and during the association with UGT1.

7) An event triggers the dissociation of UGT1 and TAPBPR from reglucosylated HLA-A, allowing it to engage with the PLC and load high affinity peptides. (Insight into the dissociation step using TAPBPR-reconstituted cells is limited since TAPBPR overexpression shifts the equilibrium towards HLA-A/TAPBPR association and favors retention in the ER).

This model explains

A) Preferential binding of HLA-A allomorphs by TAPBPR since HLA-B alleles are less stable and less likely to self-load in the ER.

B) Related and synergistic, but non-identical functions of TAPBPR and tapasin in quality control.

C) – Effect of tapasin-deficiency vs. TAPBPR-deficiency on MHC class I surface expression and stability.

Although the revised model suggests a secondary role for TAPBPR in quality control, it does not undermine the novelty or significance of this manuscript! In fact, I think it would be very interesting to discuss how the same region of tapasin and TAPBPR (N-term loop with C95 or C94) has evolved to co-opt components of ER glycan chaperone system (i.e., ERp57/ calreticulin and UGT1) to improve the stability of MHC class I complexes.

*Reviewer #3:*

In this manuscript, Boyle and colleagues present interesting and novel data that identify the glucosyltransferase UGT1 as interaction partner for the MHC-I chaperone TAPBPR which plays, in addition to tapasin, a role in the process of optimizing MHC-I peptide ligands, however, is not part of the TAP-associated peptide loading complex. It is convincingly shown that a conserved, presumably unpaired cysteine residue (C94) in TAPBPR is essential for the association of UGT1, while no evidence for the formation of intra- or intermolecular disulfide bonds (at least with heterologous molecules of appreciable molecular weight) involving C94 is found. TAPBPR[C94A] seems to be as stable as WT TAPBPR. Furthermore, it is demonstrated that the TAPBPR C94A mutation negatively influences the re-glucosylation and association with TAP1 and tapasin of HLA-A*6802 molecules, but apparently not of HLA-B*1503 molecules. Very interestingly, the peptide ligandomes of HLA-A*6802 and HLA-B*1503 molecules are shown to be specifically influenced by either TAPBPR WT of TAPBPR[C94A]. Only for HLA-A*6802 molecules, TAPBPR WT improved the loading with peptides containing classic anchors. On the other hand, the C94A mutation does not seem to reduce the capacity of TAPBPR to bind MHC-I.

The shown experiments are convincing and carefully controlled. Conclusions are valid and discussed concisely and with expertise, however, a number of important points need to be addressed satisfactorily.

Specific points:

Figure 1B: In TAPBPR precipitates from KBM-7 cells, the MHC-I WB band seems to show a clearly increased intensity in the tapasin KO situation, while this not obvious (by visual inspection) in TAPBPR precipitates from HeLa/tapasin KO cells (Figure 1A). This result may suggest competition between tapasin and TAPBPR for MHC-I binding. Please comment whether this has been more often in independent experiments, and if yes, discuss the finding. Is co-IP of MHC-I by tapasin vice versa increased after TAPBPR knock-down? Include an MHC-I blot in Figure 1C.

Figure 2E: In the size exclusion chromatography experiment, His-tagged TAPBPR WT should be compared with TAPBPR[C94A] side by side. In the SDS-PAGE insert, elution fractions should be labelled and Mw markers indicated.

Figure 3A: Indicate molecular weight markers.

Figure 4C: TAPBPR WT must be included in this peptide competition experiment.

Figure 4D: Show levels of TAPBPR WT and C94A in transfected HeLa/TAPBPR KO cells by Western blot.

Figure 4F: Spell out the classic peptide anchors that were analysed in HLA-A*6802 and HLA-B*1503 ligands.

Figure 4E-F: Can anything be said about the distribution of auxiliary anchors and preferred residues in positions other than P2 and P-Omega in peptides unique to TAPBPR WT and C94A?

Figure 5: Show densitometric analyses of the 2 bands representing MHC-I allomorphs. It seems that TAPBPR WT also somewhat increases the pulldown of HLA-B*1503 by GST-CRTwt. Where are HLA-C and HLA-E allomorphs expected to run?

Discussion, third paragraph, correct: “[…]but now it transpires[…]”

It is striking that both TAPBPR WT and C94A expression result in a significant downregulation of surface-expressed HLA-A*6802 and B*15:03 allomorphs (Figure 4D) although only A*6802 seems to have augmented peptide editing upon UGT1-mediated re-glucosylation. This is counterintuitive since the presence of the peptide editor TAPBPR should improve the overall stability of MHC-I molecules and thereby increase cell surface steady-state levels (NB: tapasin KO but not overexpression has previously been associated with reduced MHC-I cell surface levels). First experiments must be performed that address the fate of A68 and B15 in the secretory route in the absence of TABPBR and in the presence of TAPBPR WT and C94A. Since TAPBPR has no ER retention signal it is possible that the pulldown with exogenous CRT reflects a mixture of ER and post-ER MHC-I molecules. Figure 5A should include a digestion with Endo H before WB for MHC-I. Measuring the turnover of a cohort of cell-surface biotinylated MHC-I molecules in the absence of TABPBR and in the presence of TAPBPR WT and C94A would be another informative approach to clarify the effects of TAPBR-mediated peptide editing and resulting changes in the stability of MHC-I.

---

## [Author Response]

*Reviewer #1:*

*The work is done well, taking advantage of unique reagents, but my major concern is that a mechanistic explanation for the role of C94 is not clear, and is not sufficiently addressed in Discussion. In addition, a number of nagging details in the description of the reagents, in the referencing, and in experimental controls should be addressed. One issue not dealt with in the Discussion or the model is the quantitative relationships of tapasin/MHC interaction as compared with TAPBPR interaction and the relative levels of expression of tapasin and TAPBPR in different compartments and different tissues.*

Regarding the role of C94, we have now included an additional paragraph in the Discussion section to discuss some of the potential mechanistic roles of C94 in the association with UGT1. Furthermore, we have included some additional data regarding residues nearby C94 which we predict form part of a helix. We reveal mutation of these residues decreases UGT1 association with TAPBPR, suggesting C94 is part of a larger interaction face.

While the quantitative relationship between the tapasin/MHC class I interaction and the TAPBPR/MHC class I interaction, as well as the relative expression level of these proteins in different cellular compartments and in different tissues in highly interesting, this was not within the remit of this particular manuscript and it is a significant area of research we are actively working on currently.

*Reviewer #2:*

*This manuscript reports a novel interaction between TAPBPR and UGT1 and provides mechanistic explanations for previously unexplained phenomena regarding the function of TAPBPR in antigen processing. Overall, the experiments are well-designed and executed, and the manuscript is well-written. Although I generally agree with the interpretation of the data in the Results section, I have significant questions and concerns about the model (Figure 6) that is presented in the Discussion section. I acknowledge the challenge of incorporating decades of research into a model, but believe that we can generate one that is more simplistic and comprehensive of past and present data.*

We are glad you found the experiments to be well-designed and executed and that the manuscript is well-written. Furthermore, we really appreciate your insight into model we generated. In response to your suggestions we have made a significant number of changes to the model and now feel an improved model has been generated as a result.

*Issues regarding the model:*

*The difference in tapasin-dependence for HLA-A vs HLA-B alleles is a highly relevant issue, but largely ignored. HLA-A molecules are not tapasin-dependent (Greenwood 1994) and do not necessarily interact with the PLC prior to deglucosylation by GLSII (step 1). Self-loaded MHC molecules are therefore an alternate entry point into the "TAPBPR cycle". This is supported by the finding that the TAPBPR-MHC interaction is increased in the absence of tapasin (Boyle 2013). Furthermore, self-loaded HLA-A molecules are more likely to be TAPBPR substrates than PLC-loaded HLA-A molecules.*

Given that we have made an interesting observation here between the HLA-A and HLA-B molecules expressed in HeLa cells only, but have not confirmed this yet either using a large panel of different HLA molecules reconstituted into a MHC class I negative cell line or demonstrated that the HLA-A versus HLA-B effect occurs in a number of other cell lines, we feel it is premature for our model to differentiate on the effect of TAPBPR on HLA-A versus HLA-B. However, this reviewer has raised a number of very important points that we were unaware of and which we did not take into consideration when generation our initial working model, particularly regarding the point that selfloaded MHC class I molecules will be deglucosulated by GLSII in a PLC-independent manner. We thank reviewer 3 for raising this highly relevant point. Given its significance when considering the reglucosylation of MHC class I by the TAPBPR:UGT1 complex and we agree that this must be incorporated into our working model.

*The major flaw (and constraint) of this model is the authors' view that tapasin and TAPBPR are functionally equivalent in regards to peptide editing.*

*A) If true, what would determine whether TAPBPR association with HLA-A leads to step 4 (peptide exchange) vs. step 5 (peptide displacement and UGT1 association)? If true, why would a redundancy of tapasin/TAPBPR function be energetically favorable? In other words, why go through additional steps of UGT1 reglucosylation and PLC association (steps 5 and 6) if TAPBPR can do the same thing all by itself (step 4)?*

*B) Keeping in mind the number of studies on tapasin vs. TAPBPR, I think it's premature to conclude that their functional activities are identical. Although I would agree that TAPBPR stabilizes HLA-A molecules and promotes the displacement of peptides, the "selectivity" of TAPBPR activity has yet to be convincingly demonstrated (Hermann 2015, Morozov 2016, this study). More comprehensive studies are needed to directly compare tapasin and TAPBPR activities using a broader panel of low and high affinity peptides.*

We do not feel that tapasin and TAPBPR are functionally equivalent in regard to peptide editing and apologise if this has come across incorrectly in our initial manuscript. Although both molecules can function as peptide editors in vivo, we feel that the distinct cellular environment these two chaperones function in, particularly in regard to peptide available, will strongly influence whether the final outcome of these chaperone on MHC class I is peptide loading or peptide dissociation. We have made a number of changes to the Introduction and the Discussion to reflect that tapasin and TAPBPR are not functionally equivalent. Furthermore, we have also modified our model to suggest TAPBPR is more likely to enhance peptide dissociation, due to not be incorporated into the peptide loading complex.

*C) Another oversight of this model (and past studies of TAPBPR) is the peptide supply. Immunofluorescence and EndoH experiments (Boyle 2013) show that TABPBR is not exclusively localized to the ER. Even within the ER, the peptide supply may be significantly higher near TAP. Thus, the localization of TAPBPR vs. tapasin may have a profound effect on their potential to edit the peptide repertoire* in vivo.

We agree entirely with this point.

*Revised model:*

*I propose an alternative model for discussion with the authors and reviewers.*

*1) Some HLA-A molecules assemble with peptides independently of the PLC. Of these, some bind high affinity peptides and some bind suboptimal peptides.*

*2) HLA-A molecules that have bypassed the PLC are more rapidly trimmed by GlsII because the glycan is not "protected" by calreticulin. Self-loading may have occurred before or after GlsII trimming. (I do not recall evidence that would support one option vs. the other).*

*3) HLA-A molecules loaded with high affinity peptides pass quality control and do not interact with TAPBPR.*

*4) HLA-A molecules loaded with suboptimal peptides associate with TAPBPR in the ER or cis-Golgi.*

*5) Due to differences in TAPBPR vs. tapasin activity and/or the peptide supply, TAPBPR promotes the dissociation of the suboptimal ligand, but does not load a high affinity peptide.*

*6) Instead, TAPBPR improves the efficiency of reglucosylation by recruiting UGT1. Another advantage of the TAPBPR interaction is that HLA-A molecules are stabilized before and during the association with UGT1.*

*7) An event triggers the dissociation of UGT1 and TAPBPR from reglucosylated HLA-A, allowing it to engage with the PLC and load high affinity peptides. (Insight into the dissociation step using TAPBPR-reconstituted cells is limited since TAPBPR overexpression shifts the equilibrium towards HLA-A/TAPBPR association and favors retention in the ER).*

*This model explains*

A) Preferential binding of HLA-A allomorphs by TAPBPR since HLA-B alleles are less stable and less likely to self-load in the ER.

B) Related and synergistic, but non-identical functions of TAPBPR and tapasin in quality control.

*C) Effect of tapasin-deficiency vs. TAPBPR-deficiency on MHC class I surface expression and stability.*

In light of reviewer 2’s comments and suggestions we have made significant changes to our model to encompass as many of the points raises as possible. However, there is one particular point made by this reviewer which we do not feel we have data to strongly support; that TAPBPR works only on HLA-A and not HLA-B. While it is true that TAPBPR does appear to bind more strongly to HLA-A, in a cellular environment TAPBPR also affects the peptide repertoire presented on HLA-B. Therefore, we do not feel comfortable to make such a strong distinction between HLA-A and HLA-B as has been suggested. However, we are very grateful for the comments received, particularly the inclusion of self-loaded MHC class I molecule, and agree with the suggestion that TAPBPR is likely to have a more profound effect on the tapasin-independent/self-loading MHC class I molecules and have incorporated this into our working model.

*Although the revised model suggests a secondary role for TAPBPR in quality control, it does not undermine the novelty or significance of this manuscript! In fact, I think it would be very interesting to discuss how the same region of tapasin and TAPBPR (N-term loop with C95 or C94) has evolved to co-opt components of ER glycan chaperone system (i.e., ERp57/ calreticulin and UGT1) to improve the stability of MHC class I complexes.*

*Reviewer #3:*

*[…] Specific points:*

*Figure 1B: In TAPBPR precipitates from KBM-7 cells, the MHC-I WB band seems to show a clearly increased intensity in the tapasin KO situation, while this not obvious (by visual inspection) in TAPBPR precipitates from HeLa/tapasin KO cells (Figure 1A). This result may suggest competition between tapasin and TAPBPR for MHC-I binding. Please comment whether this has been more often in independent experiments, and if yes, discuss the finding. Is co-IP of MHC-I by tapasin vice versa increased after TAPBPR knock-down? Include an MHC-I blot in Figure 1C.*

In one of our previous publications (Hermann et al., J immunol, 2013, 191: 5743-5750) we found that in the absence of tapasin, the association of MHC class I with TAPBPR was increase suggesting there was some competition between the two chaperones. However, in the absence of TAPBPR, the interaction between MHC class I and tapasin did not significantly increase. Our finding here in both Figure 1B and 1A are consistent with these previously published findings. Although in Figure 1A the amount of MHC class I bound to TAPBPR in tapasin-deficient HeLa-M cells does not appear to increase compared to the tapasin-competent HeLa-M cells, western blot analysis on the lysates of these two cell lines reveals a significant loss of MHC class I molecules expression in the absence of tapasin. Therefore, there is a relative increase in the amount of MHC class I bound to TAPBPR in the absence of tapasin. We have included some additional text in the Results section and in the figure legend to this effect.

As requested we have also included a MHC class I blot in Figure 1C.

*Figure 2E: In the size exclusion chromatography experiment, His-tagged TAPBPR WT should be compared with TAPBPR[C94A] side by side. In the SDS-PAGE insert, elution fractions should be labelled and Mw markers indicated.*

As suggested we have now included the size exclusion experiment for WT-TAPBPR and the MW markers and elution fractions have been labelled.

*Figure 3A: Indicate molecular weight markers.*

The suggested changes have been made to this figure.

*Figure 4C: TAPBPR WT must be included in this peptide competition experiment.*

We have now repeated the experiment for Figure 4C an additional three independent times. As a result we have now included WT-TAPBPR into Figure 4C for comparison as requested. This demonstrates there is no change in ability of C94A-TAPBPR to discriminate between high affinity and low affinity peptide compared to WT-TAPBPR.

*Figure 4D: Show levels of TAPBPR WT and C94A in transfected HeLa/TAPBPR KO cells by Western blot.*

As these are cell lines in which the WT and C94A-TAPBPR have been stably integrated via lentiviral transduction, the level of WT and C94A expressed in these cells is the same as in Figure 3A (and also in the original Figure 5 which is now Figure 6). To make this clearer, we have included some additional text in the results referring to Figure 4D to direct the reader to Figure 3A to observe the steady state level of TAPBPR in the cell lines.

*Figure 4F: Spell out the classic peptide anchors that were analysed in HLA-A*6802 and HLA-B*1503 ligands.*

This peptide anchors for the two HLA molecules have now been included into Figure 4F (now Figure 5D).

*Figure 4E-F: Can anything be said about the distribution of auxiliary anchors and preferred residues in positions other than P2 and P-Omega in peptides unique to TAPBPR WT and C94A?*

We have now included WebLogo depiction of the 9-mer peptides isolated from cells expressing TAPBPR^WT^ and TAPBPR^C94A^ (Figure 5—figure supplement 2) which not only helps to demonstrate the distribution of anchor residues but also those at the other positions in the peptides. However, this analysis does not suggest a significant change in residues for at other positions in the peptides isolated.

*Figure 5: Show densitometric analyses of the 2 bands representing MHC-I allomorphs. It seems that TAPBPR WT also somewhat increases the pulldown of HLA-B*1503 by GST-CRTwt. Where are HLA-C and HLA-E allomorphs expected to run?*

We have now performed densitometric analyses of the 2 bands representing the MHC class I allomorphs which can be found in Figure 6—figure supplement 1. However, as this analysis has been performed on film, it is not appropriate for accurate quantification due to the signals not necessarily being on a linear scale and/or saturated as can be seen with TAPBPR^WT^ samples. Therefore to further valid the increased reglucosylation of HLA-A68 on the presence of TAPBPR^WT^ we have performed accurate quantitative analysis on tapasin immunoprecipitation in Figure 6C. In the pulldowns with GST-CRT we agree this is a slight increase in HLA-B15 associated suggesting the TAPBPR:UGT1 complex can reglucosylate HLA-B15, but the dominant change appears with HLA-A68. Furthermore, pulldown on endogenous tapasin suggests WT-TAPBPR strongly influences HLA-A68 over HLA-B15 and there is no obvious increase in HLA-B15 association with the PLC in this situation. Regarding HLA-E and –C, we have not specifically taken these into account mainly due to their lower expression relative to HLA-A and -B. Furthermore, our evidence to date suggests TAPBPR does not associate with HLA-E in HeLa cells.

*Discussion, third paragraph, correct: “[…] but now it transpires[…]”*

Transpires has been changed to appears.

*It is striking that both TAPBPR WT and C94A expression result in a significant downregulation of surface-expressed HLA-A*6802 and B*15:03 allomorphs (Figure 4D) although only A*6802 seems to have augmented peptide editing upon UGT1-mediated re-glucosylation. This is counterintuitive since the presence of the peptide editor TAPBPR should improve the overall stability of MHC-I molecules and thereby increase cell surface steady-state levels (NB: tapasin KO but not overexpression has previously been associated with reduced MHC-I cell surface levels). First experiments must be performed that address the fate of A68 and B15 in the secretory route in the absence of TABPBR and in the presence of TAPBPR WT and C94A. Since TAPBPR has no ER retention signal it is possible that the pulldown with exogenous CRT reflects a mixture of ER and post-ER MHC-I molecules. Figure 5A should include a digestion with Endo H before WB for MHC-I. Measuring the turnover of a cohort of cell-surface biotinylated MHC-I molecules in the absence of TABPBR and in the presence of TAPBPR WT and C94A would be another informative approach to clarify the effects of TAPBR-mediated peptide editing and resulting changes in the stability of MHC-I.*

As requested we have included and Endo-H digest on the exogenous CRT pulldowns. This has revealed that the MHC class I molecule which interact with the exogenous CRT are entirely Endo H sensitive and therefore, the effect of the TAPBPR:UGT1 complex on the reglucosylation of MHC class I is occurring before the medial-Golgi. This data can be found in Figure 6—figure supplement 1C.

To address the resulting changes in the stability of MHC class I, we have previously addressed the effect of TAPBPR mediated editing in the presence and absence of TAPBPR in our previous publication in *eLife* (Hermann et al. 2015). However we have provided some additional data to demonstrate the change in stability of HLA-A68 in cell expressing C94A compared to WT-TAPBPR. We have found in cells expressing C94A-TAPBPR HLA-A68 surface expression can be rescue by addition of exogenous peptide, suggesting there is a pool of peptide-receptive HLA-A68 released to the surface in these cells. Furthermore, we can recreate this phenotype on WT-TAPBPR expressing cells by depleting shUGT1 expression. We hope you find this attempt to address your query is satisfactory and informative. Unfortunately, we were unable to measure with turnover of cell surface biotinylated in the time-frame permitted for revisions.